# Inherited polygenic effects on common hematological traits influence clonal selection on *JAK2*$^{V617F}$ and the development of myeloproliferative neoplasms

Jing Guo [1,2], Klaudia Walter [1], Pedro M. Quiros[1,3,4], Muxin Gu[1],
E. Joanna Baxter[4], John Danesh[1,2,5,6,7], Emanuele Di Angelantonio[2,5,6,7,8],
David Roberts[2,9], Paola Guglielmelli [10], Claire N. Harrison[11], Anna L. Godfrey[12],
Anthony R. Green[3,13], George S. Vassiliou [3,13], Dragana Vuckovic[1,2,14,15],
Jyoti Nangalia [1,3,12,13,15] ✉ & Nicole Soranzo [1,2,6,8,13,15] ✉

Myeloproliferative neoplasms (MPNs) are chronic cancers characterized by overproduction of mature blood cells. Their causative somatic mutations, for example, *JAK2*$^{V617F}$, are common in the population, yet only a minority of carriers develop MPN. Here we show that the inherited polygenic loci that underlie common hematological traits influence *JAK2*$^{V617F}$ clonal expansion. We identify polygenic risk scores (PGSs) for monocyte count and plateletcrit as new risk factors for *JAK2*$^{V617F}$ positivity. PGSs for several hematological traits influenced the risk of different MPN subtypes, with low PGSs for two platelet traits also showing protective effects in *JAK2*$^{V617F}$ carriers, making them two to three times less likely to have essential thrombocythemia than carriers with high PGSs. We observed that extreme hematological PGSs may contribute to an MPN diagnosis in the absence of somatic driver mutations. Our study showcases how polygenic backgrounds underlying common hematological traits influence both clonal selection on somatic mutations and the subsequent phenotype of cancer.

Myeloproliferative neoplasms (MPNs) are rare chronic hematological cancers characterized by the overproduction of mature blood cells leading to elevated blood cell parameters. They are typically driven by somatically mutated *JAK2*-mediated, calreticulin (*CALR*)-mediated or *MPL*-mediated clonal expansion[1]. *JAK2* mutations are found in both polycythemia vera (PV) and essential thrombocythemia (ET), which are distinct but overlapping MPNs characterized by increased numbers of red blood cells and platelets, respectively. Mutant *JAK2* is commonly

[1]Wellcome Sanger Institute, Hinxton, UK. [2]National Institute for Health Research Blood and Transplant Research Unit in Donor Health and Genomics, University of Cambridge, Cambridge, UK. [3]Wellcome–MRC Cambridge Stem Cell Institute, Jeffrey Cheah Biomedical Centre, University of Cambridge, Cambridge, UK. [4]Instituto de Investigación Sanitaria del Principado de Asturias (ISPA), Oviedo, Spain. [5]British Heart Foundation Cardiovascular Epidemiology Unit, Department of Public Health and Primary Care, University of Cambridge, Cambridge, UK. [6]British Heart Foundation Centre of Research Excellence, University of Cambridge, Cambridge, UK. [7]Health Data Research UK Cambridge, Wellcome Genome Campus and University of Cambridge, Cambridge, UK. [8]Fondazione Human Technopole, Milan, Italy. [9]NHS Blood and Transplant–Oxford Centre, John Radcliffe Hospital and Radcliffe Department of Medicine, University of Oxford, John Radcliffe Hospital, Oxford, UK. [10]Department of Experimental and Clinical Medicine, Center for Research and Innovation of Myeloproliferative Neoplasms (CRIMM), AOU Careggi, University of Florence, Florence, Italy. [11]Department of Haematology, Guy's and St Thomas' NHS Foundation Trust, London, UK. [12]Cambridge University Hospitals NHS Trust, Cambridge, UK. [13]Department of Haematology, University of Cambridge, Cambridge, UK. [14]Department of Epidemiology and Biostatistics, School of Public Health, Faculty of Medicine, Imperial College London, London, UK. [15]These authors jointly supervised this work: Dragana Vuckovic, Jyoti Nangalia, Nicole Soranzo. ✉e-mail: jn5@sanger.ac.uk; ns6@sanger.ac.uk

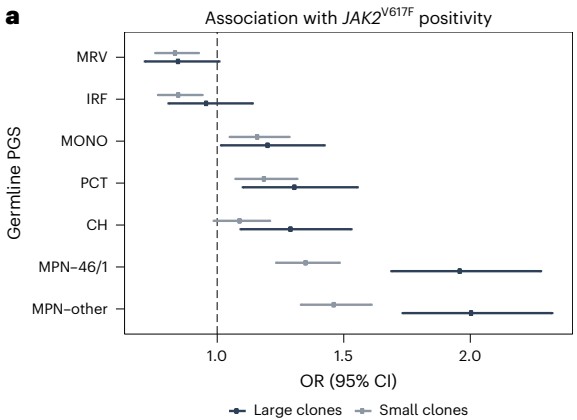

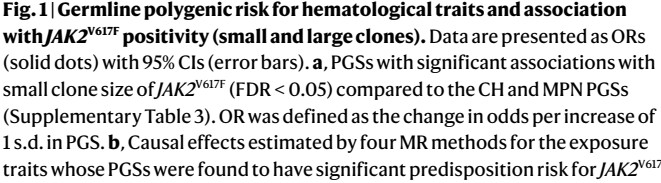

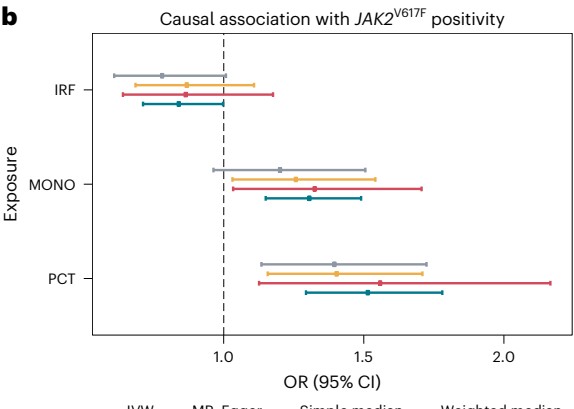

**Fig. 1 | Germline polygenic risk for hematological traits and association with $JAK2^{V617F}$ positivity (small and large clones).** Data are presented as ORs (solid dots) with 95% CIs (error bars). **a**, PGSs with significant associations with small clone size of $JAK2^{V617F}$ (FDR < 0.05) compared to the CH and MPN PGSs (Supplementary Table 3). OR was defined as the change in odds per increase of 1 s.d. in PGS. **b**, Causal effects estimated by four MR methods for the exposure traits whose PGSs were found to have significant predisposition risk for $JAK2^{V617F}$

positivity (Supplementary Table 7). OR was defined as the change in odds per increase of 1 s.d. in exposure. The MR results shown were based on GWAS summary statistics for $JAK2^{V617F}$ positivity in the full UKBB (Supplementary Fig. 4). Results based on the main discovery set (200k UKBB-WES cohort) are shown in Supplementary Table 6. The MR result for MRV was not available due to a lack of corresponding GWAS summary data in INTERVAL.

detectable in 0.1–3% of the healthy population as clonal hematopoiesis (CH)[2–7], with the vast majority of carriers not meeting or going on to develop disease-defining characteristics of MPN. Little is understood about why only a minority of individuals with mutated *JAK2* develop more severe hematological manifestations of MPN and the factors that influence blood count heterogeneity in MPNs.

The 46/1 haplotype near *JAK2* is a known germline risk factor for MPNs in the population[8]. Genome-wide association studies (GWAS) have identified additional disease-associated germline risk loci, estimating the liability-scale heritability of MPNs based on common single-nucleotide polymorphisms (SNPs) to be ~6.5% (refs. 9–11). However, these germline risk loci insufficiently explain the phenotypic heterogeneity observed within MPNs and in *JAK2*-mutated healthy carriers.

Blood cell traits vary widely in the healthy population. The genetic architecture underlying these traits is highly polygenic, with more than 11,000 independently associated genetic variants discovered so far[12–14]. These genome-wide associated variants, when combined in polygenic scores (PGSs), explain a large proportion of phenotypic variance among healthy individuals (from 2.5% for basophil count to 27.3% for mean platelet volume) and are associated with multiple common diseases and rare hematological disorders[14]. We hypothesized that a genetic burden of germline variants associated with extreme hematological traits could influence phenotypic heterogeneity in association with mutated *JAK2*, by influencing the clonal dynamics of mutant *JAK2* and/or modifying its downstream consequences. In this study, we integrate information on somatic driver mutations, germline genetic variants associated with MPNs, and CH and hematological trait PGSs to study how inherited polygenic variation underlying blood cell traits influences clonal selection on mutated *JAK2* and MPN disease phenotypes (Supplementary Fig. 1).

## Results

### Inherited polygenic contribution to $JAK2^{V617F}$ positivity

One in 30 healthy individuals reportedly harbors $JAK2^{V617F}$ in their blood, as determined using sensitive assays[6]. The majority of such individuals have low levels of $JAK2^{V617F}$ and do not meet clinical criteria for MPN due to the absence of elevated blood cell parameters. We wished to understand whether inherited polygenic loci that underlie blood cell traits influence the strength of clonal selection on $JAK2^{V617F}$.

We studied the germline characteristics of individuals in UK Biobank (UKBB) with and without $JAK2^{V617F}$. From 162,534 genetically unrelated individuals of European ancestry within the UKBB whole-exome sequencing cohort ('200k UKBB-WES cohort'; Methods), we identified 540 individuals with one or more mutant reads for $JAK2^{V617F}$ (0.3%, median variant allele frequency (VAF) = 0.056, range = 0.019–1; Supplementary Fig. 2; 'UKBB-$JAK2^{V617F}$ cohort'). The lower rate of $JAK2^{V617F}$ in the UKBB-WES cohort compared to other population studies[6,7] could be explained by its low sequencing coverage (21.5× depth), as also reported previously[15] (Supplementary Fig. 3). As expected, there was some overlap among individuals with $JAK2^{V617F}$ and those with a diagnosis of MPN. Of the 423 individuals labeled with a diagnosis of MPN (156 with ET, 161 with PV and 106 with myelofibrosis (MF)), 72 were positive for $JAK2^{V617F}$ (Supplementary Table 1).

We built PGSs for 29 blood cell traits covering a wide range of hematopoietic parameters (Supplementary Table 2). Blood cell trait-specific PGSs were then weighted (by effect size) by the sum of all common (minor allele frequency (MAF) > 0.01) variants that were independently associated with a blood cell trait at genome-wide significance ($P < 5 × 10^{-8}$) in UKBB (Methods)[14]. To assess the association between hematological PGSs and small (VAF < 0.1, $n = 397$) or large (VAF ≥ 0.1, $n = 143$) $JAK2^{V617F}$ clones, we used multinomial logistic regression including PGSs for each hematological trait (units of s.d.), together with previously reported germline sites associated with MPN[9] and CH[16] ($PGS_{MPN}$ and $PGS_{CH}$) as covariates. To account for the recognized predisposition risk for MPN driven by the *JAK2* 46/1 haplotype[8], we computed two $PGS_{MPN}$ scores, separating rs1327494 (tagging the *JAK2* 46/1 haplotype; $PGS_{MPN-46/1}$) from nontagging *JAK2* variants ($PGS_{MPN-other}$). We found a negative association between the PGSs for both mean reticulocyte volume ($PGS_{MRV}$) and immature reticulocyte fraction ($PGS_{IRF}$) and small $JAK2^{V617F}$ clones ($P = 6.2 × 10^{-4}$ and 0.0018, false discovery rate (FDR) < 0.05; Supplementary Table 3). We also found significant positive associations with small $JAK2^{V617F}$ clones for the PGSs of plateletcrit ($PGS_{PCT}$) and monocyte count ($PGS_{MONO}$) ($P = 9.5 × 10^{-4}$ and 0.0036, FDR < 0.05). Germline predisposition to high MONO and PCT values was also positively associated with large $JAK2^{V617F}$ clones at modest significance ($P = 0.033$ and 0.0022, FDR-adjusted $P = 0.31$ and 0.064; Fig. 1a). Repeating the analysis above excluding MPN cases still demonstrated a significant association between $PGS_{PCT}$ or $PGS_{MONO}$ and small $JAK2^{V617F}$ clones ($P < 0.013$, Bonferroni corrected; Supplementary Table 4),

suggesting that the inherited effects on $JAK2^{V617F}$ were not driven by the subset of MPN cases. These associations were independent of the known germline risk loci associated with MPN and CH (Supplementary Table 3). Validating these associations in the full UKBB-WES dataset ($n = 799$ and 326 for small and large clones, respectively, and $n = 338,919$ for controls), we again replicated the associations between $PGS_{PCT}$ and small $JAK2^{V617F}$ clones and between $PGS_{MONO}$ and large $JAK2^{V617F}$ clones at FDR < 0.05 (PCT: odds ratio (OR) = 1.15 (change in odds per increase of 1 s.d. in PGS), 95% confidence interval (CI) = 1.07–1.24, $P = 1.4 \times 10^{-4}$; MONO: OR = 1.20, 95% CI = 1.07–1.34, $P = 0.0014$; Supplementary Table 5).

To understand the causal relationship among these associations, we undertook Mendelian randomization (MR) analyses with GWAS estimates for the exposure (blood traits) and the outcome ($JAK2^{V617F}$ positivity; Supplementary Fig. 4) obtained from two independent sources. We used genetic instruments for hematological traits identified from UKBB, with effect size estimates from INTERVAL[17] ($n = 30,305$), an external independent cohort. MRV was excluded due to a lack of data in INTERVAL. Both PCT and MONO showed significant causality on the presence of a $JAK2^{V617F}$ clone based on inverse variance-weighted (IVW)[18] MR and demonstrated consistent effect estimates using two other MR methods (simple median and weighted median), suggesting that higher MONO and higher PCT values cause a detectable $JAK2^{V617F}$ clone (Supplementary Table 6).

Extending this analysis to the full UKBB-WES cohort ($JAK2^{V617F}$, $n = 1,125$; controls, $n = 338,919$) validated these causal associations with greater estimation accuracy (PCT: $OR_{IVW} = 1.52$, 95% CI = 1.29–1.78, $P = 3.0 \times 10^{-7}$; MONO: $OR_{IVW} = 1.3$, 95% CI = 1.15–1.49, $P = 4.6 \times 10^{-5}$; Fig. 1b and Supplementary Table 7). The IVW method of MR (Methods) assumes that the germline loci that drive MONO and PCT have no direct causal effect on driving a $JAK2^{V617F}$ clone (that is, there are no direct causal effects of the genetic instruments on the outcome). We found no evidence of pleiotropy using the MR-Egger[19] test; the estimated intercept was not significantly different from zero with $P = 0.84$ and $P = 0.90$ for PCT and MONO, respectively. The causal relationship was also significant for PCT and MONO ($P < 0.05$; Supplementary Table 7 and Supplementary Fig. 5). Additionally, the estimates were not biased by any potential pleiotropic outlier variants and were highly consistent with outlier-corrected causal estimates (Supplementary Table 7 and Methods). Lastly, to ensure the results were not confounded by the possibility that the genetic loci used as instruments for MR directly promoted the outcome (that is, $JAK2^{V617F}$ positivity), we repeated the analysis excluding genetic instruments associated with $JAK2^{V617F}$ positivity ($P_{association} < 10^{-6}$), as well as those that correlated with $JAK2^{V617F}$ variants (that is, those variants and $JAK2^{V617F}$ variants are in linkage disequilibrium (LD) $r^2 > 0.01$) or were in proximity to $JAK2^{V617F}$ variants (in the 10-Mb region centered on each variant), and found no major changes (Supplementary Table 8). Importantly, any reverse causal effect we detected for MONO and PCT was subtle and with pleiotropic effects ($P_{Egger} > 0.05$ and $P_{Egger-intercept} < 0.05$; Supplementary Table 9 and Supplementary Fig. 6).

Overall, the association results combined with MR suggest that higher PCT and MONO are causal for the presence of a $JAK2^{V617F}$ clone. This would also explain why individuals with germline predisposition to high PCT and MONO are also more likely to harbor a $JAK2^{V617F}$ clone. Given that acquisition of somatic mutations in blood is largely stochastic in healthy populations[20], our data suggest that genetically predicted PCT and MONO influence clonal selection on nascent $JAK2^{V617F}$ cells to promote mutation acquisition.

### Germline contribution to blood cell count variation in MPNs

Having shown that polygenic germline loci can predispose to $JAK2$ clone positivity through their influence on blood cell trait levels, we next studied the contribution of these inherited sites to clinical phenotypes of MPN. We first considered the four blood cell traits that are used to define MPN subcategories clinically[21] as follows: hemoglobin

concentration (HGB) (g dl$^{-1}$ divided by 10), hematocrit (HCT) (%), platelet count (PLT) ($\times 10^9$ divided by 1,000) and white blood cell count (WBC) ($\times 10^9$ divided by 100). We used SNP arrays to measure genome-wide polymorphism in an MPN cohort of 761 patients (PV, $n = 112$; ET, $n = 581$; MF, $n = 68$), in whom diagnostic blood cell counts were available and mutation status for a panel of cancer-associated genes (Fig. 2a) had previously been characterized[22].

We built PGSs for the four blood cell traits in both patients with MPN and a cohort of healthy blood donors from the INTERVAL study ($n = 30,305$; Methods). For each trait, we built a linear regression model with predictors that included the corresponding PGS (for example, $PGS_{HGB}$), demographic variables (age and sex), $JAK2$ 46/1 haplotype status (due to its influence on hematological traits[23,24]) and somatic mutation status for genes frequently (>20 patients) mutated in the MPN patient cohort (Fig. 2b). Following stepwise regression, we identified three somatic mutations ($JAK2$ mutation, $CALR$ mutation and chromosome 9 aberration resulting in homozygous $JAK2$ mutation), PGS and sex as significant explanatory variables for at least one of the four traits ($P < 0.001$, Bonferroni corrected; Fig. 2c). We then estimated the phenotypic variation in blood cell traits explained by each variable conditional on the others (Fig. 2d and Methods). As a benchmark, we estimated PGS-explained hematological trait variation in controls from the INTERVAL cohort based on similar linear regression models with covariates such as age, sex and ten principal components (PCs) controlling for population stratification. Blood cell trait phenotypes were inverse-normal transformed, and only genetically unrelated individuals were included (Supplementary Fig. 7 and Methods).

The estimated PGS-explained phenotypic variance in blood cell traits in INTERVAL was as follows: 6.8% (95% CI = 6.2–7.4%) for HGB; 6.7% (95% CI = 6.1–7.2%) for HCT; 10% (95% CI = 9.7–11%) for WBC; and 25% (95% CI = 24–26%) for PLT, highly consistent with previously published results[14]. In the MPN patient cohort, when taking into account the effects of somatic mutations, there remained significant but smaller PGS effects on HGB, HCT and WBC ($P < 0.001$, Bonferroni corrected; $n = 380$ to 577; Fig. 2c and Methods), explaining only 2.0% (95% CI = 0.62–4.2%), 3.0% (95% CI = 0.79–7.0%) and 2.8% (95% CI = 1.0–5.3%) of trait variance, respectively (Fig. 2d), while PGS had no significant effect on PLT (Fig. 2c; $P > 0.05$).

To validate these findings, we analyzed the UKBB-WES cohort including patients with MPN ('UKBB-MPN cohort'; patients, $n = 423$; healthy controls, $n = 161,872$; Supplementary Table 1) with hematological PGS and $JAK2^{V617F}$ somatic mutation as the main explanatory variables (Fig. 2e). Around 60% of MPNs were positive for $JAK2^{V617F}$. However, only 44 of 423 individuals (10.4%) with labels of ET, PV or MF in UKBB had two or more mutant reads for $JAK2^{V617F}$ at a median sequencing depth of 21× and only 72 of 423 individuals (17.2%) had one or more mutant reads for $JAK2^{V617F}$ (Supplementary Table 1), suggesting that, despite the low depth of exome sequencing coverage, this cohort of individuals may represent a mixture of those with true somatic mutation-driven MPN and those with high blood counts driven by other causes. Both PGS and $JAK2^{V617F}$ in the UKBB-MPN cohort captured a substantial proportion of the phenotypic variation for all four traits (PGS, 9.8–14.3%; $JAK2^{V617F}$, 2.8–12.0%; Methods and Fig. 2f). Interestingly, the contribution of PGS to blood cell traits was notably higher in the UKBB-MPN cohort than in the MPN patient cohort (Fig. 2d,f). This may reflect an ascertainment bias in estimation of the genetic weights of the PGSs from the UKBB. However, it is also possible that some individuals with labels of MPN included in the UKBB-MPN cohort had elevated blood counts driven by a high PGS. This could also contribute to the low prevalence of $JAK2^{V617F}$ in the UKBB-MPN cohort compared to the more strictly defined clinical MPN patient cohort.

### Germline polygenic impact on MPN subtype at diagnosis
We next explored whether polygenic loci underlying hematological traits also affect initial disease classification, severity and subsequent

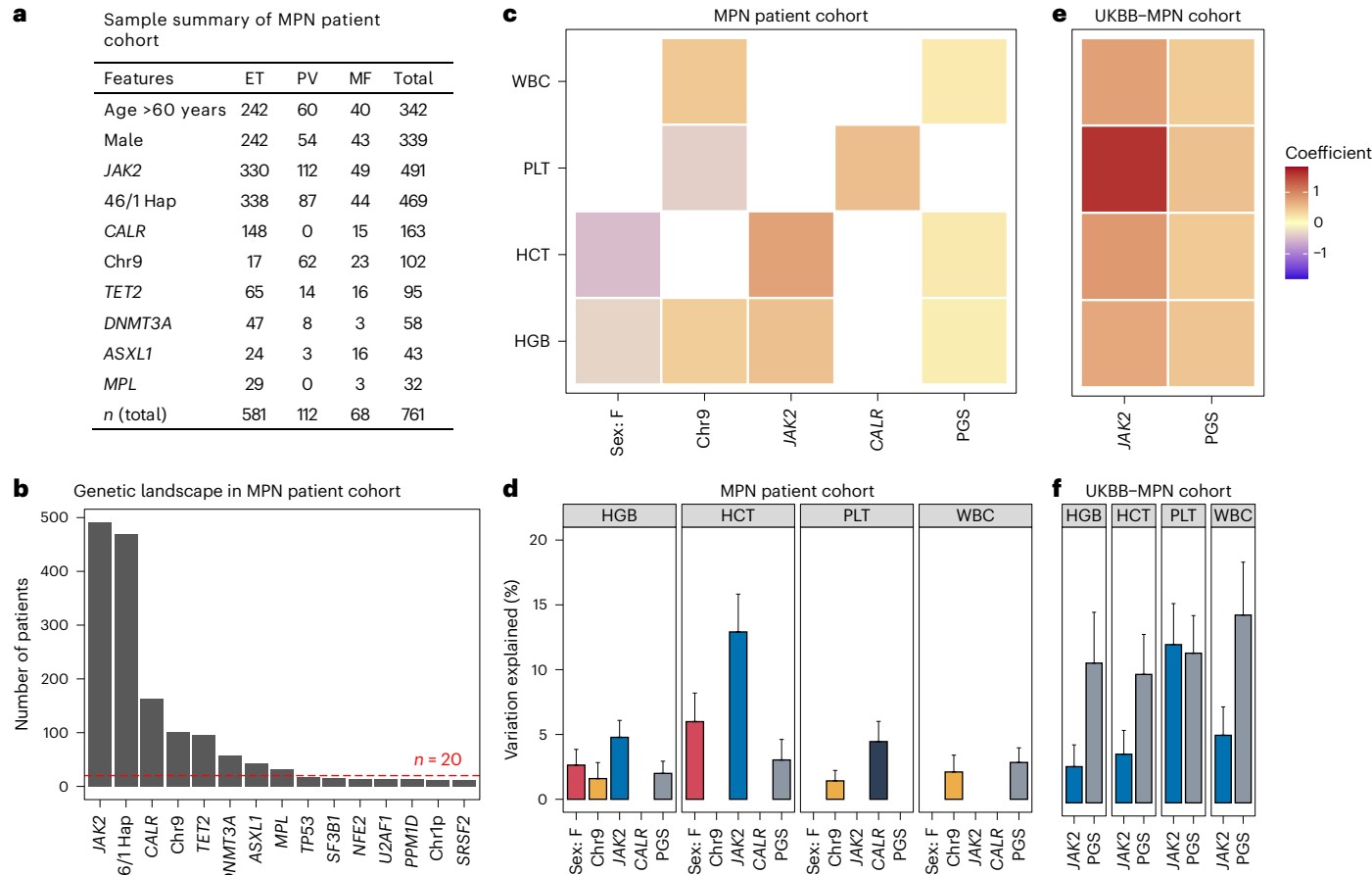

**Fig. 2 | Phenotypic variation of MPN-relevant hematological traits influenced by germline and somatic factors. a**, Table summarizing the MPN patient cohort (total *n* = 761) across demographic and genetic features (somatically mutated genes present in >20 patients). 46/1 Hap, *JAK2* 46/1 germline haplotype; Chr9, chromosomal gain or loss of heterozygosity on chromosome 9. **b**, Plot showing the distribution of germline and somatic changes in the MPN patient cohort. Somatic mutations, chromosomal aberrations and germline risk loci present in at least ten patients are shown. Chr1p, chromosomal gain or loss of heterozygosity on chromosome 1p. **c**,**e**, Regression coefficients of significant (*P* < 0.0011, Bonferroni corrected) demographic, somatic or germline variables for hematological traits in the MPN patient cohort (**c**) and *JAK2* and PGS (*P* < 0.0063, Bonferroni corrected) in the UKBB-MPN cohort (**e**). The regression coefficient (standard error (s.e.)) was estimated using linear regression analysis.

Variables shown were identified from a set of variables selected by a stepwise model selection procedure (Supplementary Tables 10 and 11 and Methods). **d**,**f**, Plots showing the estimated hematological trait variation (%; bars) with 1 s.e. (error bars) explained by the variables in **c** and **e**. The percentage variation shown for a variable was estimated conditional on the remaining variables in a selected model with the s.e. estimated via bootstrapping (Methods). Blood trait phenotype data were available for *n* = 380–577 patients in the MPN patient cohort and *n* = 268–281 individuals in the UKBB-MPN cohort (varied across traits). For a specific trait, a variable has a blank result if it did not pass the significance threshold (*P* = 0.0011 for the MPN patient cohort and 0.0063 for the UKBB-MPN cohort, Bonferroni corrected; Methods). *JAK2* includes V617F and exon 12 mutation in the MPN patient cohort and V617F only in the UKBB-MPN cohort; *CALR*, exon 9 frameshift (+1 bp) mutation.

disease evolution. MPNs can be classified into chronic phase conditions (ET and PV) and advanced phase MF. To assess whether PGSs for hematological traits influence MPN classification at diagnosis, we used multinomial logistic regression to explore the associations with standardized PGSs, including age, sex and ten PCs as covariates. We performed this analysis in genetically unrelated individuals across PGSs for the 29 different blood cell traits, including the 4 tested previously (Supplementary Table 2; ET, *n* = 581; PV, *n* = 112; control, *n* = 30,305).

We found that PGSs for multiple MPN-relevant blood traits showed significant associations with ET at FDR < 0.05. High PGS_PLT, PGS_PCT and PGS_WBC were associated with increased risk while high PGS_HCT, PGS_HGB and PGS_PDW (platelet distribution width) were associated with decreased risk of having an ET diagnosis (Supplementary Fig. 8a and Supplementary Table 12). Increased risk of PV diagnosis was modestly associated with an increased PGS for several red blood cell traits (PGS_HGB, PGS_HCT and PGS_RBC), PGS_PCT and PGSs for white blood cell traits (eosinophil count (EO) and MONO). PGS_MRV showed a risk-decreasing

effect for PV (*P* < 0.05; Supplementary Fig. 8a and Supplementary Table 12). We repeated this analysis in the UKBB-MPN cohort and healthy controls (ET, *n* = 156; PV, *n* = 161; control, *n* = ~161,000; Supplementary Table 1), taking into account the VAF of *JAK2*^V617F (Supplementary Fig. 2) because it can influence blood count parameters (Fig. 2d,f). We found that the PGSs for two platelet traits (PGS_PCT and PGS_PLT) were significant risk factors for an ET diagnosis while those for four red blood cell traits (PGS_HGB, PGS_HCT, PGS_RBC and PGS_MCHC (mean corpuscular hemoglobin concentration)) were significant risk factors for PV diagnosis at FDR < 0.05 (Supplementary Fig. 8b and Supplementary Table 13). Thus, we replicated the significant polygenic germline risk effects of PGS_PLT and PGS_PCT for ET and PGS_HGB, PGS_HCT and PGS_RBC for PV in both the MPN patient and UKBB-MPN cohorts. These results provide evidence of a strong polygenic germline predisposition for one hematological malignancy over the other, in this case ET versus PV, irrespective of somatic driver mutation status and driven by inherited variants implicated in basic hematopoietic processes.

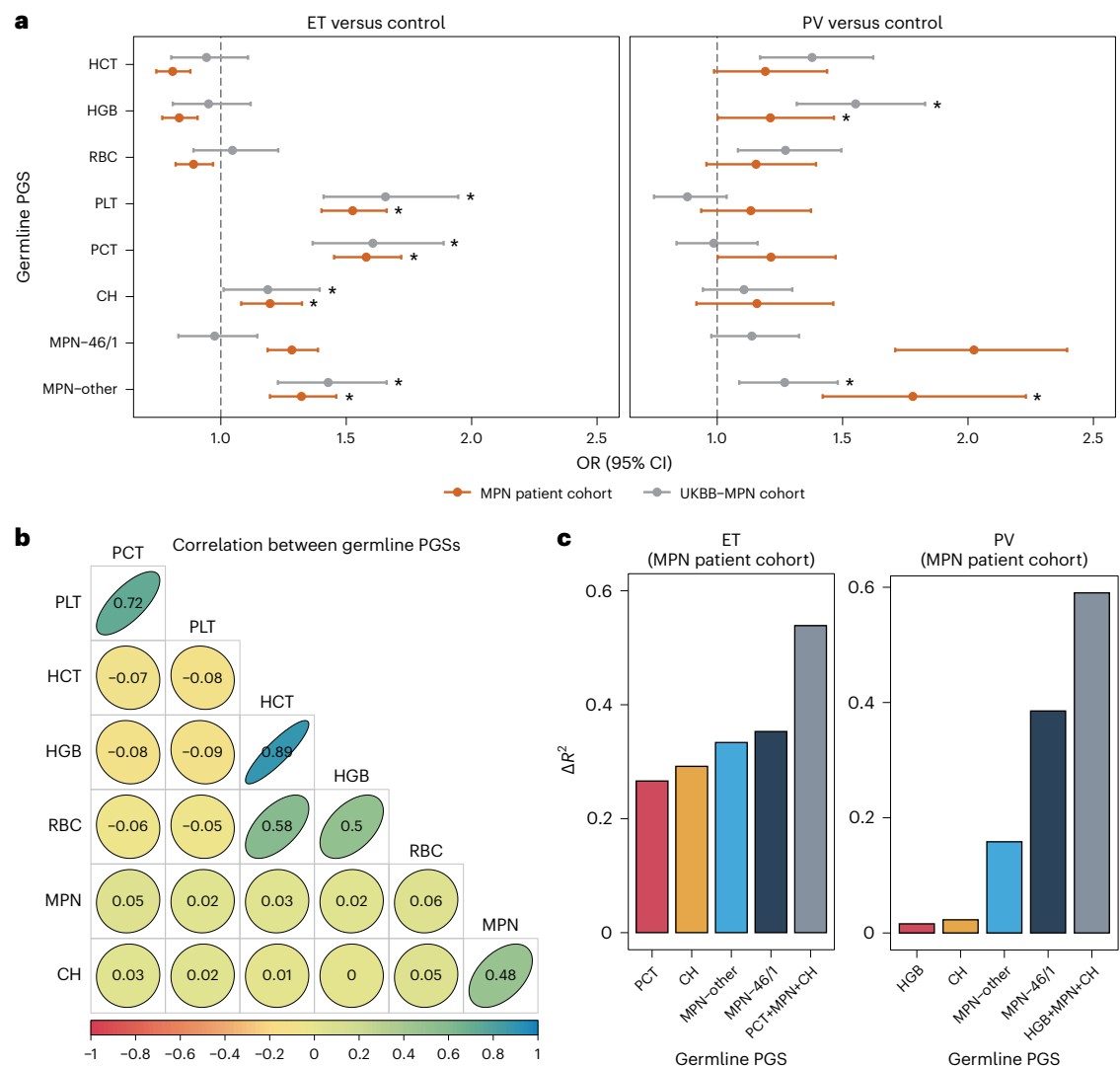

**Fig. 3 | Polygenic germline contribution of hematological traits to ET and PV relative to the germline determinants of MPN and CH. a**, The estimated OR for ET and PV across five significantly associated hematological trait PGSs, $PGS_{CH}$, $PGS_{MPN-46/1}$ and $PGS_{MPN-other}$ in the MPN patient cohort (red; ET, $n = 581$; PV, $n = 112$; control, $n = 30,305$) and the UKBB-MPN cohort (gray; ET, $n = 156$; PV, $n = 161$; control, $n = 161,872$). ORs were estimated in a multinomial logistic regression model. Data are presented as the OR (change in odds per increase of 1 s.d. in PGS; solid dots) with 95% CI (error bars). An asterisk indicates significant germline

PGSs ($P < 0.05$ in both cohorts). The corresponding ORs and $P$ values are shown in Supplementary Table 15. Age, sex and ten PCs were fit as covariates for both datasets. The VAF of $JAK2^{V617F}$ and the sample batch were two additional covariates included for the UKBB-MPN cohort. **b**, Pearson correlation coefficients across the PGSs for blood cell traits, CH and MPN in the MPN patient cohort. **c**, $\Delta R^2$ between the full and reduced models ($y$ axis) after removing each PGS component ($x$ axis). 'HGB+MPN+CH' and 'PCT+MPN+CH' represent the results when simultaneously excluding the three PGS items.

We next asked whether an individual such as a $JAK2^{V617F}$ carrier might be protected from developing an MPN by inheriting a low PGS for relevant blood cell traits. Using enrichment tests in the full UKBB-WES cohort across the PGSs of six hematological traits that were identified to be either putative causal factors for $JAK2^{V617F}$ clones or associated factors for MPN diagnosis (MONO, PCT, PLT, HGB, HCT and RBC; Methods), we found that healthy $JAK2^{V617F}$ carriers were enriched in the low-PGS group for the two platelet traits and monocytes, with an enrichment OR around 2 ($PGS_{PCT}$: OR = 2.8, 95% CI = 1.51–5.42, $P = 3.8 \times 10^{-4}$; $PGS_{PLT}$: OR = 2.35, 95% CI = 1.29–4.43, $P = 0.0027$; $PGS_{MONO}$: OR = 1.99, 95% CI = 1.11–3.67, $P = 0.015$), indicating a protective effect that makes low-PGS individuals around two times less likely to have ET than those in the high-PGS group. This is interpreted as a relative risk, which is very close to OR given the low incidence of ET (~1.6 per 100,000; Supplementary Table 14). Importantly, this indicates that an individual's PGS for several hematological traits also influences the risk of developing

subsequent disease from $JAK2^{V617F}$ CH. The association of low PGS with healthy $JAK2^{V617F}$ carriers was also confirmed in a logistic regression analysis ($PGS_{PCT}$: OR = 2.32, 95% CI = 1.27–4.25, $P = 0.0065$; $PGS_{PLT}$: OR = 2.48, 95% CI = 1.36–4.54, $P = 0.0032$; $PGS_{MONO}$: OR = 2.08, 95% CI = 1.15–3.77, $P = 0.016$) with covariates included (Methods).

Of note, only 10–17% of the UKBB-PV cohort had a $JAK2^{V617F}$ mutation ($n = 1$ or $\geq 2$ reads, respectively; Supplementary Table 1), although mutated $JAK2$ is expected to be found in >99% of PV cases. This raises the possibility that polygenic germline predisposition to high red blood cell indices may also contribute to other causes of clinical polycythemia not driven by $JAK2^{V617F}$ mutation[25]. However, we cannot exclude the possibility that some $JAK2$ mutations were missed due to the length of time between UKBB blood sampling and diagnosis (Supplementary Fig. 9) and the low sequencing coverage of $JAK2$, although these factors would equally affect the PV subgroups positive and negative for $JAK2^{V617F}$ in the UKBB cohort (Supplementary Fig. 3).

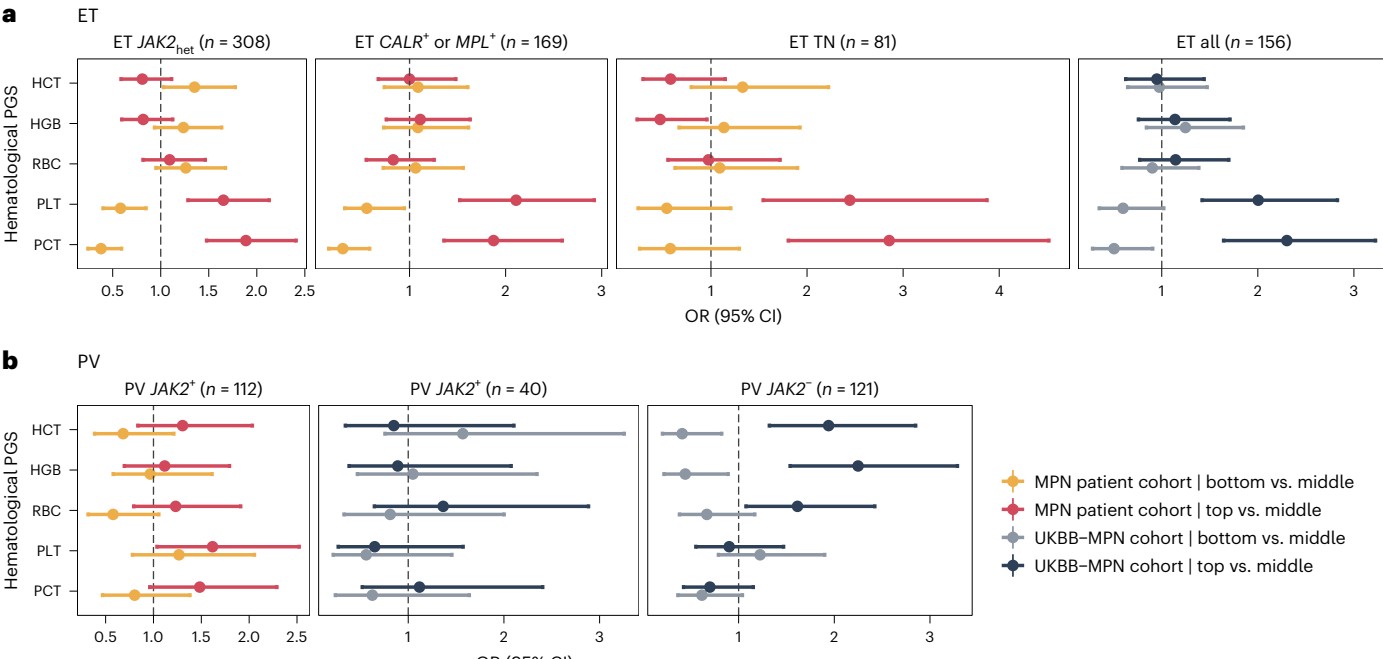

**Fig. 4 | Germline polygenic risk for ET and PV in the context of somatic mutations. a,b,** ORs of the top and bottom PGS quintiles relative to ORs of the middle three quintiles in the MPN patient cohort (red and yellow) and the UKBB-MPN cohort (blue and gray) for ET (**a**) and PV (**b**). Data are presented as the OR (change in odds per increase of 1 s.d. in PGS; solid dots) with the 95% CI (error bars). Mutation-stratified groups are shown wherever these data were available. $n$ is the sample size of ET (or PV) cases in a specific group. Notably, PV negative for *JAK2* mutation should be a rare diagnosis in PV (normally <5%), but several such individuals were present in the UKBB-WES cohort ($n = 121$ of 161; Supplementary Table 1). It may be that many of these individuals do not have an underlying

clonal disorder such as PV but present with secondary erythrocytosis, due, for example, to smoking, alcohol or lung disease, where a germline predisposition to high HGB or HCT may contribute to blood count phenotypes that mimic PV (Supplementary Fig. 11). There was limited statistical power to demonstrate any significance for the PV subgroup positive for *JAK2* mutation given the small sample size (UKBB-MPN cohort, $n = 40$). Furthermore, we cannot exclude the possibility that a mutant *JAK2* clone was missed during sequencing in some of these individuals due to the low sequencing depth in UKBB or the timing of the blood sample relative to diagnosis.

## Combined germline impact on MPN classification at diagnosis

Because several SNPs and germline loci have been found to be associated with both MPN and CH[9,15,26–29], we assessed whether germline predisposition to an ET versus PV diagnosis through PGSs for blood cell traits was independent of previously reported germline risk loci. To this end, we estimated the independent germline effects of each of the five hematological traits significant to MPN (for example, $PGS_{PCT}$), taking into account previously reported genetic loci associated with MPN[9] and CH[16] ($PGS_{MPN-46/1}$, $PGS_{MPN-other}$ and $PGS_{CH}$). Across both MPN patient cohort and UKBB-MPN cohort, the strongest ($P < 0.05$ in both cohorts) germline risk factors for a diagnosis of ET were $PGS_{PLT}$ and $PGS_{PCT}$, followed by MPN-specific risk loci (that is, $PGS_{MPN-other}$). In PV, $PGS_{HGB}$ and $PGS_{MPN-other}$ were the strongest risk factors (Fig. 3a, Supplementary Table 15 and Methods). Our data confirmed strong risk effects for these five hematological PGSs independent of all currently known genetic loci predisposing to risk of MPN and CH. As a sensitivity analysis, we confirmed these associations after excluding variants associated with MPN and CH and their proxies (LD $r^2 > 0.6$) from the hematological PGSs (Methods). Interestingly, the 46/1 haplotype, which was most strongly associated with PV in the MPN patient cohort, was not significant for PV in the less well-defined UKBB-PV cohort (Fig. 3a), in which the majority of individuals were $JAK2^{V617F}$ negative, suggesting that this locus is not a risk factor for developing high red blood cell indices independently of mutant *JAK2*, such as in secondary or apparent polycythemia.

Next, we fit the same variables as above in a linear regression model and quantified the contribution of each of the germline components (that is, $PGS_{MPN-46/1}$, $PGS_{MPN-other}$, $PGS_{CH}$ and PGS of a blood cell trait) to ET or PV as the relative difference in the variance explained ($R^2$) on the liability scale between a full model and a reduced model excluding one

or more variables ($\Delta R^2 (\%) = \frac{R^2_{full} - R^2_{reduced}}{R^2_{full}} (\%)$; Methods). We selected one representative hematological PGS for each analysis because the PGS correlations were high within trait categories (for example, Pearson's $r = 0.72$ between $PGS_{PLT}$ and $PGS_{PCT}$) and very small between trait categories (for example, $r = -0.08$ between $PGS_{PLT}$ and $PGS_{HCT}$; Fig. 3b). Thus, we selected $PGS_{PCT}$ for ET and $PGS_{HGB}$ for PV, both of which showed the strongest disease predisposition risk. Consistent with the pattern of estimated risk (Fig. 3a), the germline contribution (relative to the full model-explained variation) of PCT polygenic loci to ET (26.6%) was largely comparable to the germline components of CH (29.2%) and MPN (33.4% for $PGS_{MPN-other}$ and 35.3% for $PGS_{MPN-46/1}$). For PV, germline loci associated with MPN (15.8% for $PGS_{MPN-other}$ and 38.5% for $PGS_{MPN-46/1}$) were more dominant than polygenic loci for HGB (1.6% for $PGS_{HGB}$; Fig. 3c). For both ET and PV, more than 50% of the variance explained by the full model was accounted for by using combined germline components, highlighting the strong heritable predisposition to specific MPN subtypes. We also explored germline associations with MF presentation and MF transformation, but we did not see any significant associations that replicated across both the MPN patient and UKBB-MPN cohorts.

## Influence of hematological PGSs on somatic mutation subtypes of MPN

Finally, we explored whether the influence of PGSs for blood cell trait variation on MPN classification was altered by the different somatic driver mutations underlying MPN. We stratified patients with ET according to their somatic driver mutation into patients heterozygous for *JAK2* mutation ($JAK2_{het}$; that is, *JAK2* mutation without chromosome

9 aberrations; $n = 308$), patients with *CALR* and/or *MPL* mutations ($n = 169$), and patients negative for *JAK2*, *MPL* and *CALR* mutations (triple-negative (TN) patients; $n = 81$). We then compared diagnoses for patients in either the top or bottom quintile of the PGS distribution to those for patients in the middle quintiles (Methods). Our results showed consistent associations of both $PGS_{PLT}$ and $PGS_{PCT}$ with ET ($n = 581$) regardless of the driver mutation (Fig. 4a). Indeed, the top quintiles for $PGS_{PLT}$ and $PGS_{PCT}$ were associated with higher odds of an ET diagnosis, even in individuals without somatic driver mutations (ET TN, $n = 81$), suggesting that such individuals may simply represent those with extreme PGSs for PLT and PCT. We replicated these results in the UKBB-MPN cohort (Fig. 4a).

For patients with PV in the MPN patient cohort, we did not see significant differentiation between the top and bottom quintiles for blood cell trait PGSs, possibly because the mutant *JAK2* clone dominates the PV phenotype in this patient cohort (PV *JAK2*$^+$ including *JAK2*$_{het}$ and *JAK2*$_{hom}$ in Fig. 4b; $n = 112$), resulting in the PGSs not having an observed impact. A similar pattern was also present in individuals with PV positive for *JAK2* mutation in the UKBB-MPN cohort, although the number of such individuals was small ($n = 40$) as the overall UKBB-PV cohort had much higher than expected rates of PV negative for *JAK2* mutation. Despite similar observations for patients in both the UKBB-MPN and MPN patient cohorts, we cannot rule out the possibility that we had limited statistical power to discover any significance for blood cell trait PGSs. This is especially true for the UKBB-MPN cohort, which had only 40 individuals with PV carrying *JAK2*$^{V617F}$, as we noted that the higher PGS quintiles (for example, the fourth and fifth quintiles) for $PGS_{HGB}$, $PGS_{HCT}$ and $PGS_{RBC}$ tended to have more patients with PV than the lower quintiles, and similarly so for $PGS_{PLT}$ and $PGS_{PCT}$ for patients with ET (Supplementary Fig. 10). In the UKBB-MPN cohort, we consistently observed that the top PGS quintile of $PGS_{HGB}$, $PGS_{HCT}$ and $PGS_{RBC}$ presented risk effects for a PV label without *JAK2*$^{V617F}$ ($n = 121$; Fig. 4b). We did not see any significant associations between the PGSs for blood cell traits and progression of disease to MF that replicated in both cohorts, consistent with the notion that disease transformation is driven more by the acquisition of additional somatic driver mutations.

## Discussion

Germline variants have been shown to contribute to CH[26,28–33], as well as blood cancers such as MPNs[8–10]. However, the influence of polygenic inherited variation controlling normal hematopoiesis on the consequences of somatic driver mutations in blood and the development of blood cancers has not previously been explored.

Recent data suggest that acquisition of driver mutations in blood is much more frequent than previously appreciated[20,34]. In the presence of a high stochastic acquisition rate of driver mutations, variation in the presence of a detectable mutant *JAK2* clone could reflect differential selection landscapes among individuals. Indeed, *JAK2* somatic mutations generally occur long before the onset of MPNs[35,36], and heterogeneity in the strength of clonal selection on mutant *JAK2* is evident in both healthy individuals[37] and those with MPN[35]. The factors driving such differences in selection remain unclear. We find that high $PGS_{MONO}$ and high $PGS_{PCT}$ are associated with detectable clone positivity for *JAK2*$^{V617F}$. Our forward MR analyses suggest that the mode of action is via a route where the elevated blood counts themselves positively select on mutant *JAK2*. Indeed, clonal expansion of nascent mutated hematopoietic stem cells in MPNs and CH has been suggested to be influenced by inflammation[38–41]. However, we caution that a sensitivity analysis distinguishing causality from potential genetic correlation should be considered in future studies[42], although no significant genetic correlation was detected in our data. We note that the causal relationship inferred by MR does not necessarily suggest a direct association between baseline MONO or PCT levels themselves and positivity for *JAK2*$^{V617F}$, as blood cell parameters can suffer from measurement error and/or high levels of variability due to unknown or unmeasured environmental factors. These factors can counteract genetic effects, the latter of which provide better proxies for stable long-term exposure of a trait on an outcome.

In patients with MPN, we showed that germline variants affecting platelets ($PGS_{PCT}$ and $PGS_{PLT}$) and HGB ($PGS_{HGB}$ and $PGS_{HCT}$) significantly increased the risk of developing ET or PV independently of somatically mutated clones. This may be because some individuals, through their polygenic germline risk, have higher or lower baseline levels of blood count traits, which causes them to meet the diagnostic criteria for a particular MPN disease subtype[21] sooner. However, it is also conceivable that somatic driver mutations result in differential downstream consequences in the presence of a high germline predisposition to one versus another hematological trait. Importantly, our data highlight how extreme PGSs for hematological traits may contribute to individuals meeting the diagnostic criteria for MPN in the absence of somatic mutations, potentially resulting in erroneous labels of MPN or 'triple-negative' MPN. This emphasizes the clinical need to consider whether somatic mutation-negative individuals with elevated blood cell counts suspicious of MPN, currently around 10% of patients, truly have underlying clonal neoplastic disease.

We explored the reasons for low rates of *JAK2*$^{V617F}$ positivity in the UKBB-MPN cohort. We found that nearly half the individuals with PV were diagnosed after a blood draw in UKBB, potentially explaining the low rate of positivity for *JAK2* mutation (Supplementary Fig. 9). Furthermore, the overall depth of sequencing for *JAK2* was low, as previously reported[15], compared to other CH-related genes. However, these factors would have similarly affected the PV groups positive and negative for *JAK2*$^{V617F}$ (Supplementary Figs. 3 and 9) and therefore should not unduly bias the results for germline polygenic risk in the context of somatic mutations in the UKBB-MPN cohort, where a significant differentiation in polygenic effects between the top and bottom quintiles of PGSs was observed in PV cases that were not carriers of *JAK2*$^{V617F}$ compared to those that were carriers (Fig. 4b). This suggests that polygenic germline predisposition contributes to a PV diagnostic label in *JAK2*$^{V617F}$-negative individuals in UKBB and should be considered as a potential contributory factor in *JAK2*-unmutated individuals with high HGB or HCT.

In summary, by analyzing two large MPN disease cohorts and UKBB, we provide new insights into the interaction between germline polygenic variation involved in basic hematopoiesis and clonal selection on somatic driver mutations in blood and describe how this interaction can influence the phenotype of subsequent blood cancer. Our results highlight an independent and causal new component of the overall susceptibility to clonal disease and provide a new framework for considering an individual's genetic background in the context of their clinical presentation.

## Online content

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

## Methods

### Samples and consent

UKBB analyses were undertaken under application numbers 56844 and 13745. Samples from patients with MPN were obtained following written informed consent and ethics approval as described previously[22]. Briefly, samples from patients with MPN were collected from outpatient clinics at Addenbrooke's Hospital, Guy's and St Thomas' Hospital in the UK, under the clauses of the 'Causes of Clonal Haematological Disorders Project', which had regional ethical approval from the Eastern Multi-region Ethics Committee (MREC 02/5/22 and 07/MRE05/44) and local research and ethical approval at participating UK hospitals. Additional MPN samples were obtained from the University of Florence Careggi Hospital, Italy, with local ethics approval. Whole blood-derived samples were additionally analyzed from the Primary Thrombocythaemia-1 (PT1) trials, a multicenter international trial in ET. Analyses for this study were conducted under Cambridge Blood and Stem Cell Biobank ethics, 18/EE/0199 expiry 14 July 2024.

### Genotyping, quality control and imputation

We genotyped peripheral blood-derived DNA extracted from whole blood or granulocyte samples from a cohort of 1,358 patients with MPN[22] ('MPN patient cohort') using the Affymetrix UKBB array ($m$ = ~731,000 loci). For the raw genotype data, we removed samples with dish quality control (QC) <0.82 (a default threshold of a measure of the extent to which the distribution of signal values separates from background values), sample call rate <97% (an estimate of the overall quality for a sample), plate pass rate <98% and average QC plate pass rate for the remaining samples for a given plate <99%. We used the program apt-genotype-axiom to make genotyping calls. We excluded samples with an outlying heterozygosity rate of greater than ±3 s.d. and high identity-by-descent sharing of >0.9 to obtain 1,207 samples. Genotype QC for the control cohort INTERVAL ($n$ = ~44,000) was as previously published[12]. We performed PC analysis in a combined dataset of MPN cases, INTERVAL samples and the 1000 Genomes Project (1000G) with major global populations. We removed individuals more than 5 s.d. from the mean of the British ancestry group (GBR-1000G) to obtain 1,010 patients with MPN and 30,949 control individuals from INTERVAL. For genotype imputation, we included autosomal variants with an SNP call rate of >99%, MAF of >0.01, and Hardy–Weinberg equilibrium $P > 10^{-6}$ in controls ($m$ = ~523,000) and $P > 10^{-10}$ in patients with MPN ($m$ = ~576,000). We imputed genotype data to the HRC v1.1 reference panel using the Michigan imputation server (minimac4)[43]. We extracted the SNPs with an imputation accuracy $R^2$ value of >0.6, MAF of >0.01, and Hardy–Weinberg equilibrium $P > 10^{-6}$ for controls and $P > 10^{-10}$ for cases ($m$ = ~7.6 million) in unrelated individuals with genetic relatedness of <0.05 for both patients with MPN ($n$ = 980) and controls in INTERVAL ($n$ = 30,630). A genetic relationship matrix (GRM) was built on genotyped variants after LD pruning (LD $r^2$ < 0.2; $m$ = ~215,000) in the combined case–control dataset. For UKBB, genotype imputation, variant QC, and sample filtering of ancestry outliers and genetic relatedness of individuals were as previously described[44]. The corresponding approved application number was 13745.

### Inclusion criteria for MPN and $JAK2^{V617F}$ positivity

The MPN disease cohort had detailed somatic mutation status for 69 myeloid cancer-associated genes, clinical phenotypes of four blood cell traits (HGB and HCT levels and platelet and white blood cell counts) at diagnosis, plus age, sex and information on disease transformation. UKBB-WES[44] data from 200,450 individuals (200k UKBB-WES as our main discovery dataset) were used to identify known MPN cases through annotations for ET (ICD-10 D47.3, D75.2), PV (ICD-10 D45), MF (ICD-10 D47.4, D75.81), chronic myeloid leukemia (CML; ICD-10 C921, C922, C931), and chronic myeloproliferative disease (CMD; ICD-10 D47.1). For both the MPN patient cohort and the UKBB-MPN cohort ($n$ = 761 and 423, respectively, with genotype data available),

we included only patients with MPN who were explicitly diagnosed as having ET, PV or MF. Patients with conflicting records/unclassified/ other MPN (for example, CML) were excluded.

Note that the MPN cases in the UKBB-MPN cohort were not restricted to participants who had existing diagnoses at the time of blood sampling. Any participants whose inpatient records could be matched to the ICD-10 codes for MPN were included as cases in this study. To calculate the time interval from the blood draw to MPN diagnosis, we matched MPN cases to information regarding the date a blood sample was collected and when a particular diagnosis was first recorded in the hospital data (episode start date), downloaded from the UKBB data portal. When an MPN case could be matched to multiple episode start dates, the earliest date was selected as the diagnosis date. We then calculated the time interval between the two dates for ET and PV.

UKBB-WES data were also used to identify a set of individuals ($n$ = 540, of whom 72 had a corresponding diagnosis of ET, PV or MF, 6 had a diagnosis of CMD, 3 had a diagnosis of CML, 63 had more than one record of MPN subtypes and 396 were healthy) with mutant reads (either 1 or ≥2; $n$ = 359 and $n$ = 181, respectively) corresponding to the $JAK2^{V617F}$ mutation ('UKBB-$JAK2^{V617F}$ cohort'); these patients comprised those with and without a diagnosis of MPN. The mpileup function of Samtools 1.9 was used with the FASTA file of the GRCh38 assembly and the parameter '-r chr9:5073767-5073775' to calculate the number of mutant reads and coverage of each base around the V617 hotspot. We excluded reads with base quality of <13 using the base quality filter in the Samtools mpileup tool. $JAK2^{V617F}$ clone size was measured using VAF in the UKBB-WES cohort. Descriptions of the 200k UKBB-WES dataset can be found in Kar et al.[15].

### Germline associations of hematological traits with $JAK2^{V617F}$ positivity

To detect germline associations with $JAK2^{V617F}$ positivity, we classified the UKBB-$JAK2^{V617F}$ cohort as individuals with either small or large clones (VAF cut-off = 0.1) and applied multinomial logistic regression analysis (small, $n$ = 397; large, $n$ = 143 (regardless of whether the individual had MPN) versus $n$ = ~162,000 UKBB controls). The corresponding approved application number was 56844. $PGS_{MPN-46/1}$, $PGS_{MPN-other}$ and $PGS_{CH}$ along with age, sex, WES batch and ten PCs (controlling for population stratification) were fitted as covariates. The significance threshold was FDR < 0.05. The ORs (95% CIs) of $PGS_{MPN-46/1}$, $PGS_{MPN-other}$ and $PGS_{CH}$ were estimated in the same model without fitting hematological PGSs. Hematological PGSs were sourced from a published GWAS conducted in UKBB ($n$ = 408,112)[18]. SNPs associated with MPN and CH were discovered by Bao et al.[9] ($m$ = 25 with association $P < 10^{-6}$) and Kessler et al.[16] ($m$ = 57 with $P < 5 \times 10^{-8}$), respectively. We adopted the suggestive significance threshold ($P < 10^{-6}$) to extract MPN-associated SNPs to obtain variants that are potentially functionally important to MPN[9]. Only SNPs that overlapped with our common SNPs (MAF > 0.01) were incorporated in $PGS_{MPN}$ and $PGS_{CH}$ (UKBB-MPN cohort, $m$ = 25; UKBB controls, $m$ = 51). The PGS in the regression analysis was standardized with units of s.d., and an OR estimate indicates the change in odds per increase of 1 s.d. in the PGS.

We applied MR methods to identify causal associations of blood cell traits with $JAK2^{V617F}$ positivity in the UKBB-$JAK2^{V617F}$ cohort. For $JAK2^{V617F}$ positivity (≥1 mutant reads) as the outcome, we focused on the traits that showed significant $JAK2^{V617F}$ associations as exposures. Summary statistics of $JAK2^{V617F}$ positivity were obtained by conducting a case–control GWAS in unrelated individuals in the 200k UKBB-WES cohort (controls, $n$ = 161,994; cases, $n$ = 540) and also in the full UKBB-WES cohort (controls, $n$ = 338,919; cases, $n$ = 1,125) using SAIGE (v0.38)[45]. A GRM was built on variants for which imputation, QC and LD pruning with an $r^2$ threshold of 0.2 had been performed ($m$ = 341,000; MAF > 0.01). Age, sex, WES batch and ten PCs were fit as covariates. This GWAS had sample overlap with the UKBB where the GWAS of hematological traits was done. Thus, to maximize power and minimize bias

due to overfitting, we selected independent genetic instruments of blood cell traits from UKBB[14] but re-estimated weights and s.e. from an external source (INTERVAL; LD $r^2 < 0.05$; $P < 5 \times 10^{-8}$ in UKBB and <0.05 in INTERVAL). For these traits, we estimated genetic causal effects using four main MR methods implemented in mr_allmethods() in the MendelianRandomization package[46], namely, simple median-based MR[47], weighted median-based MR, IVW MR[46] and MR-Egger[19]. We preliminarily excluded variants that had a large difference in allele frequency (>0.2) between the GWAS summary data and the reference sample (the 10,000 random unrelated individuals in the UKBB-EUR set). We then selected independent common variants (MAF > 0.01; LD $r^2 < 0.05$) that were genome-wide significant in UKBB-EUR ($P < 5 \times 10^{-8}$) and nominally significant in INTERVAL ($P < 0.05$; GWAS summary data sourced from Vuckovic et al.[14]) as genetic instruments ($n = 373–703$ depending on the trait). Only blood cell traits that had GWAS summary data available in INTERVAL were included in the MR analysis. A result was identified as significant if it had $P_{IVW} < 0.0042 (= 0.05/(3 \times 4)$, Bonferroni corrected), consistent directions of the estimated causal effects across the four main methods and an MR-Egger intercept not significantly different from 0 ($P_{intercept} > 0.05$). We tested whether there was significant distortion in the causal estimate before and after removing outliers (if any) using a published method called MR pleiotropy residual sum and outlier (MR-PRESSO)[48]. MR-PRESSO detected one outlier horizontal pleiotropic variant for MONO and nine for PCT, but the distortion test before and after removing outliers was not significant for either MONO or PCT ($P_{distortion} = 0.91$ and 0.48, respectively). MR-PRESSO and MR-Egger regression complement each other in providing a less biased causal estimate with better precision: the MR-PRESSO outlier test assumes that more than half of the instruments are valid (with no horizontal pleiotropy) and have balanced pleiotropy, while MR-Egger, which corrects for the global average pleiotropy effect among all variants, is best suited when the percentage of horizontal pleiotropic variants is large (>50%)[48]. We also examined whether there was reverse causality using independent genome-wide-significant variants for $JAK2^{V617F}$ positivity as genetic instruments ($P < 10^{-6}$ (a less stringent threshold with more instruments) and LD ($r^2 < 0.05$; $m = 11$). The corresponding genetic effects of hematological traits were extracted from the GWAS in INTERVAL.

## PGS computation for blood cell traits

We obtained published summary statistics for participants of European ancestry ($n = {\sim}408,000$) in UKBB for 29 blood cell traits[14]. We refined the GWAS associations for each trait using GCTA-COJO[49] (v1.93.3 beta) with $P_{association} < 5 \times 10^{-8}$ in a 10-Mb window. GCTA-COJO selects SNPs based on conditional $P_{association}$ values through a genome-wide stepwise selection procedure and estimates the joint effects of all selected SNPs after model optimization. We randomly sampled 10,000 unrelated individuals (relatedness <0.05) of European ancestry from the UKBB as the LD reference for COJO analysis. We then computed PGSs in the MPN patient cohort ($n = 761$) and controls in INTERVAL ($n = 30,305$) using the selected SNPs based on the joint effects estimated using COJO ($m = 179–1,093$ depending on the trait). PGSs for the 29 blood cell traits in the UKBB were obtained from published data[14]. Only common SNPs (MAF > 0.01) that passed the after-imputation QC and were in common among datasets (GWAS summary data, reference data and individual-level data; $m = {\sim}7.5$ million) contributed to COJO and the PGSs. $PGS_{MPN}$ and $PGS_{CH}$ in the MPN patient cohort and INTERVAL were based on the SNPs discovered by Bao et al.[9] (19 variants with $P_{association} < 10^{-6}$ and MAF > 0.01) and Kessler et al.[26] (49 variants with association $P < 5 \times 10^{-8}$ and MAF > 0.01).

## Hematological trait variation explained by germline and somatic genetic features in patients with MPN

We studied four hematological traits available in our MPN patient cohort: HGB (g dl$^{-1}$ divided by 10), HCT (%), WBC (×10$^9$ divided by 100)

and PLT (×10$^9$ divided by 1,000). To estimate the phenotypic variation explained by a variable, we built a linear regression model with the hematological PGS, a set of driver genes, age and sex. We focused on the eight genes for which mutations were carried by more than 20 patients with MPN. We used this model as an initial model and selected the model that explained the greatest amount of phenotypic variation with the fewest independent explanatory variables. Model selection was carried out using the function stepAIC() in the package MASS (v7.3-54)[50]. Based on the selected model, we estimated the phenotypic variation explained by a variable conditional on remaining variables using the function Anova (type = 2) in the car package (v3.0-10)[51]. The s.e. was the square root of the sampling variance of the estimated proportion of explained variation computed following bootstrapping (boot() in the R package boot (v1.3-28) with the default 1,000 replications)[52,53]. A variable was considered as significant at $P < 0.0011 = 0.05/(4 \times (8 + 3))$ (PGS, age and sex that are relevant to blood cell counts and eight germline and somatic genetic changes) across four traits. We performed the analyses for ET and PV separately. For a comparison with healthy individuals, we estimated PGS-explained phenotypic variation using the same approach in the control dataset (INTERVAL) with age, sex and ten PCs as covariates. The PGSs were standardized and the phenotypes for the four traits in both the MPN patient cohort and INTERVAL controls were inverse-normal transformed. The raw phenotypes in the MPN patient cohort were preliminarily adjusted for subcohorts wherever necessary. The raw data of phenotypes in INTERVAL controls were processed and described as previously published[14]. We replicated the analysis in the UKBB-MPN cohort ($n$ between 268 and 281 due to varied data availability across traits) and UKBB controls ($n = {\sim}140,000$ to ${\sim}146,000$), based on a model with PGS and $JAK2^{V617F}$ mutation status as explanatory variables (age, sex and batch as covariates). The $P_{association}$ cut-off was 0.0063 = 0.05/(4 × 2). The PGSs based on common SNPs (MAF > 0.01) and phenotype data for the UKBB participants were as previously published[14]. Raw phenotypes were processed from a published study[12], including an adjustment for sex and environmental factors and inverse-normal transformation.

## Germline associations of blood cell traits with MPN

In the MPN patient cohort, we studied the germline associations with three aspects of MPN diagnosis: subtype classification (ET versus PV), initial severity (ET + PV versus primary MF) and transformation (primary MF versus transformed MF). For subtype classification, we applied a multinomial logistic regression model to the patients with ET or PV and controls with age, sex, ten PCs and 29 hematological PGSs (standardized). The same model was applied to the association analysis for initial severity and transformation. The significance threshold of germline associations was FDR-adjusted $P < 0.05$. For replication, the same model was applied to the UKBB-MPN cohort for subtype classification (ET versus PV) and severity (ET + PV versus MF), with the VAF of $JAK2^{V617F}$ and sample batch fit as two additional covariates. Note that it is unknown whether patients with MF in the UKBB-MPN cohort had primary MF or their disease had transformed from ET or PV.

For significant associations that were replicated in both datasets, we first quantified the germline contribution of a blood cell trait (for example, $PGS_{PCT}$) to ET and PV by fitting the same multivariate logistic regression model plus $PGS_{MPN-46/1}$, $PGS_{MPN-other}$ and $PGS_{CH}$ (and VAF of $JAK2^{V617F}$ in the UKBB-MPN cohort) as covariates. The germline effects of $PGS_{MPN-46/1}$, $PGS_{MPN-other}$ and $PGS_{CH}$ were estimated from the model without fitting a hematological PGS. We also repeated the analysis excluding the hematological variants that overlapped or were in LD ($r^2 > 0.6$) with MPN and CH risk loci.

We performed an enrichment analysis to test whether there was a polygenic germline protective effect on MPN in $JAK2^{V617F}$ carriers in the full UKBB-MPN cohort. For this, we defined high-PGS and low-PGS groups by the median of the PGS distribution for a hematological trait within the $JAK2^{V617F}$ carriers and tested whether healthy individuals were

enriched in the low-PGS group for ET and PV separately. We considered PGSs for the traits significantly associated with MPN classification (HCT, HGB, RBC, PLT and PCT) and those that were causally associated with $JAK2^{V617F}$ positivity (PCT and MONO). We estimated the PGS effect based on a logistic regression model within $JAK2^{V617F}$ carriers, where healthy carriers were coded as cases and carriers with ET or PV were coded as controls. PGS was fitted as a factor with the low-PGS group coded as 1. $PGS_{MPN-46/1}$, $PGS_{MPN-other}$, $PGS_{CH}$, age, sex, WES batch and ten PCs were fit as covariates.

### Contributions of germline components to MPN classification

We quantified the contribution of each of the germline components ($PGS_{MPN-46/1}$, $PGS_{MPN-other}$, $PGS_{CH}$ and PGS of a blood cell trait) to ET or PV by comparing the full model for a disease (for example, ET) with a reduced model excluding one or more variables. We quantified the difference in variance explained ($R^2$) between the two models, defined as $\Delta R^2 (\%) = \frac{R^2_{full} - R^2_{reduced}}{R^2_{full}} (\%)$, where the $R^2$ component (for example, $R^2_{full}$) was the variance explained on the liability scale transformed from the observed scale under-reported disease prevalence. Instead of modeling disease status on an observation scale (unaffected or affected), the liability threshold model describes disease liability on an unobserved continuous scale, assuming the sum of environmental and additive genetic components from an independent normal distribution[54]. Individuals are affected if liability exceeds a truncation threshold value. We adopted a published transformation method where the variance explained $R^2$ on a liability scale was a function of $R^2$ on an observed scale ($R^2_{full}$ and $R^2_{reduced}$), population disease prevalence (that is, 9 per 100,000 for ET and 5.4 per 100,000 for PV[55]), proportion of cases in the ascertained sample (581/(581 + 30,305) for ET and 112/(112 + 30,305) for PV in the MPN patient cohort) and height of the standard normal probability density function at the truncation threshold (equation (2) in Zhang et al.[56]).

### Germline impact in MPN influenced by somatic driver mutation

For significant germline associations that were replicated in both datasets, we performed mutation-stratified analyses focusing on $JAK2$, $CALR$ and $MPL$. The stratification resulted in four ET groups and three PV groups, which included the following: a $JAK2_{het}$ group ($JAK2^+C9^-MPL^-CALR^-$; $n = 308$), a $CALR^+$ and/or $MPL^+$ group ($JAK2^-C9^-MPL^+CALR^-$, $JAK2^-C9^-MPL^-CALR^+$ and $JAK2^-C9^-MPL^+CALR^+$; $n = 169$) and a TN group ($JAK2^-C9^-MPL^-CALR^-$; $n = 81$) in the MPN patient cohort and one ET-all group in the UKBB-MPN cohort as a replicate for ET and a PV-all group ($JAK2^+$ including 50 $JAK2_{het}$ and 62 $JAK2_{hom}$) in the MPN patient cohort and $JAK^+$ PV and $JAK2^-$ PV groups in the UKBB-MPN cohort for PV. For an MPN subtype (for example, ET), we combined the patients in a specific stratified group (for example, ET $JAK2_{het}$; $n = 308$) from the MPN patient cohort together with controls from INTERVAL and then divided the combined individuals into groups according to PGS quintiles and estimated the PGS effect on the disease in the top or bottom quintile versus the middle quintiles via logistic regression analysis with age, sex and ten PCs as covariates.

### Reporting summary

Further information on research design is available in the Nature Portfolio Reporting Summary linked to this article.

### Data availability

GWAS summary statistics for the 29 blood cell traits were sourced from a published study (Vuckovic et al.[14]) and are available from the GWAS Catalog (https://www.ebi.ac.uk/gwas/) with accession numbers GCST90002379–GCST90002407. Individual-level phenotype/genotype data for the UKBB can be requested by application at https://www.ukbiobank.ac.uk. Individual-level genotype data for the 1000 Genomes Project are available at https://www.internationalgenome.org/. INTERVAL data can be requested by application to the study leaders (https://www.intervalstudy.org.uk/), and MPN patient cohort data can be requested from the authors.

### Code availability

The case–control GWAS for $JAK2^{V617F}$ positivity in the UKBB-WES cohort was conducted using SAIGE version 0.38 (https://github.com/weizhouUMICH/SAIGE). GRM computation for the case–control GWAS for $JAK2^{V617F}$ positivity and conditional analysis for the GWAS associations for 29 blood cell traits were carried out using GCTA-GRM (https://yanglab.westlake.edu.cn/software/gcta/#MakingaGRM) and GCTA-COJO (https://yanglab.westlake.edu.cn/software/gcta/#COJO), respectively, in GCTA version 1.93.3beta2 (https://yanglab.westlake.edu.cn/software/gcta/#Download). PGS computation in the MPN patient cohort and INTERVAL, LD-based variant pruning and LD $r^2$ computation were conducted in PLINK version 1.9 (https://www.cog-genomics.org/plink/). MR analysis for $JAK2^{V617F}$ positivity was performed using the R package MendelianRandomization version 0.5.1 (https://cran.r-project.org/src/contrib/Archive/MendelianRandomization/). Hematological trait variation explained by germline and somatic genetic features in patients with MPN was estimated in R package MASS version 7.3-57 (https://cran.r-project.org/src/contrib/Archive/MASS/), car version 3.0-10 (https://cran.r-project.org/src/contrib/Archive/car/) and boot version 1.3-28 fv.

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

## Acknowledgements

J.N. is a Cancer Research UK (CRUK) advanced clinician scientist fellow. Work in the Nangalia laboratory is supported by CRUK, Wellcome core funding, and the Alborada Trust and Rosetrees Trust. D.V. is a member of the health protection research unit in chemical and radiation threats and hazards, a partnership between Public Health England and Imperial College London, which is funded by the National Institute for Health Research (NIHR). J.D. holds a British Heart Foundation professorship and an NIHR senior investigator award. Samples from patients with MPN were provided by the Cambridge Blood and Stem Cell Biobank, which is supported by the Cambridge NIHR Biomedical Research Centre, the Wellcome Trust–MRC Stem Cell Institute and the Cambridge Experimental Cancer Medicine Centre, UK. We thank Q. Zhang, N. Williams and D. Leongamornlert at the Wellcome Sanger Institute (Cambridge, UK) and N. Pirastu at the Fondazione Human Technopole (Milan, Italy) for their thoughtful discussions and constructive advice.

## Author contributions

J.G. performed all data analyses and prepared figures under the supervision of N.S. and J.N. K.W. supported data preparation. P.M.Q., M.G. and G.S.V. analyzed somatic mutations in the UKBB. E.J.B. provided MPN samples. J.D., E.D.A. and D.R. shared cohort data from INTERVAL. P.G., C.N.H., A.L.G., A.R.G. and J.N. provided MPN data and samples. D.V., J.N. and N.S. supervised the work. J.G., D.V., N.S. and J.N. wrote the paper. All authors reviewed the paper. The study was conducted in accordance with the journal's guidelines on inclusion and ethics in global research.

## Competing interests

The authors declare no competing interests.

## Additional information

**Correspondence and requests for materials** should be addressed to Jyoti Nangalia or Nicole Soranzo.

Jyoti Nangalia
Nicole Soranzo

# Reporting Summary

## Statistics

For all statistical analyses, confirm that the following items are present in the figure legend, table legend, main text, or Methods section.

| n/a | Confirmed | |
|---|---|---|
| ☐ | ☒ | The exact sample size ($n$) for each experimental group/condition, given as a discrete number and unit of measurement |
| ☐ | ☒ | A statement on whether measurements were taken from distinct samples or whether the same sample was measured repeatedly |
| ☐ | ☒ | The statistical test(s) used AND whether they are one- or two-sided *Only common tests should be described solely by name; describe more complex techniques in the Methods section.* |
| ☐ | ☒ | A description of all covariates tested |
| ☐ | ☒ | A description of any assumptions or corrections, such as tests of normality and adjustment for multiple comparisons |
| ☐ | ☒ | A full description of the statistical parameters including central tendency (e.g. means) or other basic estimates (e.g. regression coefficient) AND variation (e.g. standard deviation) or associated estimates of uncertainty (e.g. confidence intervals) |
| ☐ | ☒ | For null hypothesis testing, the test statistic (e.g. $F$, $t$, $r$) with confidence intervals, effect sizes, degrees of freedom and $P$ value noted *Give P values as exact values whenever suitable.* |
| ☒ | ☐ | For Bayesian analysis, information on the choice of priors and Markov chain Monte Carlo settings |
| ☒ | ☐ | For hierarchical and complex designs, identification of the appropriate level for tests and full reporting of outcomes |
| ☐ | ☒ | Estimates of effect sizes (e.g. Cohen's $d$, Pearson's $r$), indicating how they were calculated |

*Our web collection on statistics for biologists contains articles on many of the points above.*

## Software and code

Policy information about availability of computer code

| Data collection | We did not collect individual-level data. All datasets used in the study were sourced from public available data (eg UKBB) or from data previously collected for published studies. <br><br> MPN patients data originated from Grinfeld et al. NEJM 2018 with somatic mutation data available for each individual. Clinical data were shared by the authors. <br> UKBB-WES data were obtained from UK Biobank repository. INTERVAL data were shared by authors from the INTERVAL cohort. |
|---|---|
| Data analysis | The case-control GWAS for JAK2V617F positivity in the UKBB-WES was conducted using SAIGE version 0.38 (https://github.com/weizhouUMICH/SAIGE). GRM computation for the case-control GWAS for JAK2V617F positivity and conditional analysis for the GWAS associations for 29 blood cell traits were carried out using GCTA-GRM (https://yanglab.westlake.edu.cn/software/gcta/#MakingaGRM) and GCTA-COJO (https://yanglab.westlake.edu.cn/software/gcta/#COJO) respectively in GCTA version 1.93.3beta2 (https://yanglab.westlake.edu.cn/software/gcta/#Download). PGS computation in MPN-patient cohort and INTERVAL, LD-based variant pruning and LD r2 computation were conducted in PLINK version 1.9 (https://www.cog-genomics.org/plink/). MR analysis for JAK2V617F positivity were performed using the R package MendelianRandomization version 0.5.1 (https://cran.r-project.org/src/contrib/Archive/MendelianRandomization/). Haematological traits variation explained by germline and somatic genetic features in MPN patients were estimated in R package MASS version 7.3-57 (https://cran.r-project.org/src/contrib/Archive/MASS/), car version 3.0-10 (https://cran.r-project.org/src/contrib/Archive/car/) and boot version 1.3-28 (https://cran.r-project.org/src/contrib/Archive/boot/). |

For manuscripts utilizing custom algorithms or software that are central to the research but not yet described in published literature, software must be made available to editors and reviewers. We strongly encourage code deposition in a community repository (e.g. GitHub). See the Nature Portfolio guidelines for submitting code & software for further information.

# Data

Policy information about availability of data

All manuscripts must include a data availability statement. This statement should provide the following information, where applicable:
- Accession codes, unique identifiers, or web links for publicly available datasets
- A description of any restrictions on data availability
- For clinical datasets or third party data, please ensure that the statement adheres to our policy

GWAS summary statistics for the 29 blood cell traits were sourced from a published study (Vuckovic et al. Cell [2020]) and are available at GWAS Catalog (https://www.ebi.ac.uk/gwas/) with accession numbers GCST90002379–GCST90002407. Individual-level phenotype/genotype data of UK Biobank can be requested via application to https://www.ukbiobank.ac.uk. Individual-level genotype data of the 1000 Genomes Project are available at https://www.internationalgenome.org/. INTERVAL data can be requested via application to the study leaders (https://www.intervalstudy.org.uk/) and MPN-patient cohort data can be requested from the authors.

# Human research participants

Policy information about studies involving human research participants and Sex and Gender in Research.

| | |
|---|---|
| Reporting on sex and gender | Only sex was considered in the study with sex information determined based on individual-level genotype data using the sex check tool implemented in PLINK v1.9 (https://www.cog-genomics.org/plink/).<br>No sex-specific analysis were performed but sex as a covariate was always fitted in the model for identifying any potential sex effect. |
| Population characteristics | Age and sex:<br>In the 200k UKBB-WES data (n=162,534), 46% individuals are aged at greater than 60 years (n=74,759) and 48% are males (n=74,759).<br>In the MPN-patient cohort (n=761), 45% are aged at >60 years (n=342) and 45% are males (n=339).<br>In the healthy cohort INTERVAL (n=30,305), the median age is 43.3 with only 13% are aged at >60 years (n=3965); 50% are males (n=15,089).<br><br>Diagnosis:<br>In the 200k UKBB-WES data (n=162,534), 46% individuals are aged at greater than 60 years (n=74,759), 48% are males (n=74,759) and 423 individuals have a diagnosis of MPN including 156 ET, 161 PV and 106 MF. We identified known MPN cases through annotations for essential thrombocythemia (ET; ICD10 D47.3, D75.2), polycythaemia (PV; ICD10 D45), myelofibrosis (MF; ICD10 D47.4, D75.81), chronic myeloid leukaemia (CML; ICD10 C921, C922, C931), and chronic myeloproliferative disease (CMD; ICD10 D47.1).<br>In the MPN-patient cohort (n=761), 45% are aged at >60 years (n=342), 45% are males (n=339). All the individuals in this cohort are MPN patients including 581 ET, 112 PV and 68 MF.<br>For both the MPN-patient cohort and the MPN cases in UKBB-WES ("UKBB-MPN cohort"), we only included MPN patients who were explicitly diagnosed as ET, PV or MF; patients with conflicting records/unclassified/other MPN (e.g., chronic myeloid leukaemia), were excluded (n=761 and 423 respectively with genotype data available).<br><br>Time of the diagnosis and blood sampling of UKBB - MPN cohort:<br>The MPN cases in the UKBB-MPN cohort were not restricted to participants who had existing diagnoses at the time of blood sampling. Any participant whose inpatient records could be matched to the MPN ICD10 codes were included as cases in this study. To calculate the time interval from blood draw to MPN diagnosis, we matched MPN cases to the information regarding the date a blood sample collected and when a particular diagnosis was first recorded in the hospital data (episode start date) downloaded from the UKBB Data Portal. If an MPN case could be matched to multiple episode start dates, the earliest date was selected as the diagnosis date. We then calculated the time interval between the two dates for ET and PV respectively.<br><br>Genetic relatedness and ancestry:<br>Throughout the study, we only include genetically unrelated individuals and those that are of European ancestry. Please see below Data Exclusions for detailed inclusion cut-off in the raw data processing within this study. |
| Recruitment | UK Biobank analyses were undertaken under application numbers 56844 and 13745. MPN patient samples were obtained following written informed consent and ethics approval as described previously22. Briefly, patient samples with MPN were collected from outpatient clinics at Addenbrooke's Hospital, Guys and St Thomas' Hospital in the UK, under the clauses of the 'Causes of Clonal Haematological Disorders Project' which had regional ethical approval from the Eastern Multi-region Ethics Committee (MREC 02/5/22 and 07/MRE05/44) and local research and ethical approval at participating UK hospitals. Additional MPN samples were obtained and the University of Florence Careggi Hospital, Italy with local ethics approval. Whole blood derived samples were additionally analysed from the Primary Thrombocythaemia -1 (PT1) trials. PT1 is a multi-center international trial in ET. Analyses for this study was conducted under the Cambridge Blood and Stem Cell Biobank ethics, 18/EE/0199 expiry 14/07/2024. |
| Ethics oversight | Ethics oversight was provided by the Cambridge Stem Cell Biobank and also ethical board review at Wellcome Sanger Institute prior to sample sequencing. |

Note that full information on the approval of the study protocol must also be provided in the manuscript.

# Field-specific reporting

Please select the one below that is the best fit for your research. If you are not sure, read the appropriate sections before making your selection.

☒ Life sciences ☐ Behavioural & social sciences ☐ Ecological, evolutionary & environmental sciences

For a reference copy of the document with all sections, see nature.com/documents/nr-reporting-summary-flat.pdf

# Life sciences study design

All studies must disclose on these points even when the disclosure is negative.

| Sample size | No sample-size calculation was performed.<br><br>In the discovery analysis for the JAK2-V617F positivity, we used the 200k UKBB-WES data sourced from a published study Kar et al. Nat Genet 2022 which included all the UKBB participants for whom whole exome sequencing data have been released in December 2020.<br><br>The disease classification analysis were based on the discovery dataset we called "MPN-patient cohort". We included all the MPN-patient data sourced from a published study Grinfeld et al, NEJM 2018 only excluding those that did not pass genotyping/imputation QC, genetic relatedness and ancestry (see below). We then replicate the findings in the 200k UKBB-WES data obtained from the UK Biobank repository.<br><br>Below are the exact sample sizes in each dataset:<br>UKBB-JAK2-V617F cohort (n=540) with non-carriers in the 200k UKBB-WES (n=161,994)<br>JAK2-V617F positivity in full UKBB-WES (n=1,125 cases; 338,919 controls)<br>MPN-patient cohort (n=761) with healthy controls in INTERVAL (n=30,305)<br>UKBB-MPN cohort (n=423) with healthy controls in the 200k UKBB-WES (n=161,872) |
|---|---|
| Data exclusions | For both the MPN-patient cohort and the MPN cases in UKBB-WES ("UKBB-MPN cohort"), we only included MPN patients who were explicitly diagnosed as ET, PV or MF; patients with conflicting records, unclassified or other MPN, such as chronic myeloid leukaemia, were excluded (n = 761 and 423 respectively with genotype data available). UKBB-WES data was also used to identify a set of individuals (n = 540, of which 72 had a corresponding diagnosis of ET, PV or MF, 6 CMD, 3 CML, 63 with more than one records of MPN subtypes, and 396 healthy) with #mutant reads (either at least 1 or =>2; n = 359 and 181 respectively) corresponding to the JAK2V617F mutation ("UKBB-JAK2V617F cohort") – these patients comprised both those with and without a formal diagnosis of MPN in UKBB. The mpileup function of Samtools 1.9 was used with the FASTA file of GRCh38 assembly and the parameter "-r chr9:5073767-5073775" to calculate the number of mutant reads and coverage of each base around the V617 hotspot. We excluded reads with base quality <13 using the base quality filter in Samtools mpileup tool. JAK2V617F clone size was measured by VAF in the UKBB-WES cohort. Genotype data inclusion criteria are described above.<br><br>We excluded samples in the MPN-patient cohort (n=1,358) with an outlying heterozygosity rate >+/- 3 s.d. and high IBD sharing >0.9 and obtained 1,207 samples. In the combined dataset of MPN cases, INTERVAL and 1000 Genomes Project (1000G) with global major populations, we removed individuals >5 s.d. from the mean of the British ancestry (GBR-1000G) and obtained 1,010 MPN patients and 30,949 control individuals from INTERVAL. |
| Replication | All the Results were first described in MPN-patient cohort (with INTERVAL cohort as healthy controls) and then replicated in the UKBB-MPN cohort (with UKBB healthy as controls).We can't replicate the germline association with JAK2V617F positivity in the UKBB-JAK2V617F cohort (with UKBB-WES non carriers as controls) in MPN-patient cohort because of the unavailability of the measurements on JAK2 mutations in the control dataset INTERVAL. |
| Randomization | No randomization was applied. For identifying any germline associations, we fit age, sex, PCs (to control for potential population stratification), sample batch and VAF of JAK2V617F as covariates wherever applicable. |
| Blinding | No blinding was undertaken. |

# Reporting for specific materials, systems and methods

We require information from authors about some types of materials, experimental systems and methods used in many studies. Here, indicate whether each material, system or method listed is relevant to your study. If you are not sure if a list item applies to your research, read the appropriate section before selecting a response.

## Materials & experimental systems

| n/a | Involved in the study |
|-----|----------------------|
| ☒ | Antibodies |
| ☒ | Eukaryotic cell lines |
| ☒ | Palaeontology and archaeology |
| ☒ | Animals and other organisms |
| ☒ | Clinical data |
| ☒ | Dual use research of concern |

## Methods

| n/a | Involved in the study |
|-----|----------------------|
| ☒ | ChIP-seq |
| ☒ | Flow cytometry |
| ☒ | MRI-based neuroimaging |

