## [Peer Review File · Nature Genetics]

Peer Review Information

Manuscript Title: Inherited polygenic effects on common haematological traits influence clonal selection on JAK2-V617F and the development of myeloproliferative neoplasms

Corresponding author name(s): Dr Jyoti Nangalia, Professor Nicole Soranzo

Reviewer Comments & Decisions:

Decision Letter, initial version:
--

13th February 2023

Dear Jyoti,

Your Article "Polygenic germline risk of common haematological traits drives clonal selection on JAK2-V617F and the development of myeloproliferative neoplasms" has been seen by two referees. You will see from their comments below that, while they find your work of interest, they have raised several relevant points. We are interested in the possibility of publishing your study in Nature Genetics, but we would like to consider your response to these points in the form of a revised manuscript before we make a final decision on publication.

To guide the scope of the revisions, the editors discuss the referee reports in detail within the team, including with the chief editor, with a view to identifying key priorities that should be addressed in revision, and sometimes overruling referee requests that are deemed beyond the scope of the current study. In this case, we ask that you address all technical queries related to the study design and analytical strategy, extending the analyses where feasible and revising the presentation and interpretation of the results in light of the referees' comments. We hope you will find this prioritized set of referee points to be useful when revising your study. Please do not hesitate to get in touch if you would like to discuss these issues further.

We therefore invite you to revise your manuscript taking into account all reviewer and editor comments. Please highlight all changes in the manuscript text file. At this stage we will need you to upload a copy of the manuscript in MS Word .docx or similar editable format.

*2) If you have not done so already, please begin to revise your manuscript so that it conforms to our Article format instructions, available [here](http://www.nature.com/ng/authors/article_types/index.html). Refer also to any guidelines provided in this letter.

[redacted]

We hope to receive your revised manuscript within 8-12 weeks. If you cannot send it within this time, please let us know.

Nature Genetics is committed to improving transparency in authorship. As part of our efforts in this direction, we are now requesting that all authors identified as 'corresponding author' on published papers create and link their Open Researcher and Contributor Identifier (ORCID) with their account on the Manuscript Tracking System (MTS), prior to acceptance. ORCID helps the scientific community achieve unambiguous attribution of all scholarly contributions. You can create and link your ORCID from the home page of the MTS by clicking on 'Modify my Springer Nature account'. For more information, please visit www.springernature.com/orcid.

Sincerely,
Kyle

Kyle Vogan, PhD
Senior Editor
Nature Genetics
<https://orcid.org/0000-0001-9565-9665>

Referee expertise:

Referee #1: Genetics, cancer, myeloproliferative neoplasms

Referee #2: Genetics, complex traits, clonal hematopoiesis

Reviewers' Comments:

Reviewer #1:
Remarks to the Author:

This paper by Guo and colleagues investigates the relationship between polygenic scores (PGS) for hematologic traits and clonal selection of JAK2V617F as well as subsequent development of myeloproliferative neoplasms (MPNs). PGS for monocyte count and plateletcrit were associated with JAK2 clonal expansion, while PGS for red cell traits were associated with PV and platelet trait PGS were associated with ET. Interestingly, extreme blood cell trait PGS may account for some false diagnoses of triple negative MPNs. The manuscript is well-written, employs appropriate methods and contributes novel etiologic understanding to MPN susceptibility. Below are comments for further consideration.

- Results (end of paragraph 1) mention studying "clonal dynamics". As this study investigates JAK2-V617F at one timepoint, I suggest rephrasing as longitudinal data is not available on clonal trajectories.

- More details surrounding JAK2-V617F mutation detection in UK Biobank WES data would be helpful. What was the distribution for the sequencing coverage depth in this region? What caller(s) were used for calling JAK2-V617F? Were any exclusion criteria used to exclude samples with low coverage or remove reads with low base quality/mapping quality? As the frequency of JAK2-V617F was low in UK Biobank MPN cases, it is critical to detail the detection approach and outline limitations/thresholds of detection to put the resulting findings in context.

- It appears that MPN cases were included in the analyses on JAK2-V617F presence and clone size. Did the authors perform any stratified/sensitivity analyses among MPN-free individuals to ensure effects of PGSs on JAK2 were not driven by the subset of MPN cases?

- Some additional comment on the change in the magnitude of associations of the monocyte count and plateletcrit PGS by large vs. small clone size could be helpful, rather than focusing on significant vs. non-significant associations. Such dose response relationships are helpful for establishing causality.

- "Importantly we did not detect significant reverse causality from JAK2-V617F positivity to either PCT or MONO" – I commend the authors for performing the two-sample MR to assess directionality of effect. That said, I am curious what proportion of variance of the blood cell traits as well as JAK2-V617F was explained by the instruments used? Could a poor instrument for JAK2-V617F result in limited ability to identify reverse causality? Additionally, was JAK2 46/1 haplotype information utilized in the instrument?
- It does not appear the authors investigated how actual phenotypic measures of blood cell traits are related to JAK2-V617F positivity and clone size. Such analyses would help support the PGS findings and would provide further evidence for the "alternative" explanation for the PGS associations presented in Discussion paragraph 2. This should be possible as phenotypic measurements of blood cell counts are available in UK Biobank.
- The Figure 1 legend is missing information on abbreviations used in the figure.
- For the analyses of PGS impact on MPN subtype at diagnosis, myelofibrosis results are missing. I suspect this was a sample size issue and therefore myelofibrosis analyses were not conducted, but if myelofibrosis analyses were performed and the results were null this should be stated.
- Also for the MPN subtype associations, it was nice to see replication in UK Biobank MPN cases; however, a comparison of effect estimates between the two cohorts would be interesting, especially since UK Biobank had a much lower frequency of JAK2-V617F mutations among MPN cases.
- It was not clear in the somatic driver mutation section how JAK2-V617F carriers with chr 9 alterations were included in the analysis. I suspect a sizeable fraction of cases included both point mutations and chromosomal alterations.

Reviewer #2:
Remarks to the Author:

This manuscript explores connections between blood cell trait polygenic scores (PGS) and myeloproliferative neoplasm (MPN)-related phenotypes. Specifically, the authors consider JAK2 V617F clones in UK Biobank, MPN patients from a separate cohort, and MPN cases reported in UK Biobank. They identify associations between PGS and all of these traits, leading them to draw two main conclusions: (i) higher platelet and monocyte counts causally drive JAK2 V617F clonality; and (ii) polygenic predisposition to blood cell traits influences risk of MPNs.

The latter analyses of MPNs constitute the bulk of the paper and seem generally thorough and well-supported, with several different analyses exploring variance explained by PGSs in healthy individuals versus MPN cases, independence of PGSs from known MPN risk variants, and robustness of effects across MPN somatic driver mutation classes and across cohorts. The idea that common-variant predisposition to higher or lower blood cell indices can substantially influence risk and/or clinical classification of MPNs (for which somatic mutations are usually the focus) seems quite interesting to me -- though I should caveat that I am not a hematologist and may not be up to date with literature in this field. I did find one other recent report of blood PGSs associated with acute lymphoblastic leukemia (PMID 34469753), and I noticed that Bao et al. (ref. 9) did observe genetic correlations

between MPN risk and blood traits, so this idea is not entirely unprecedented.

I was less convinced by the Mendelian randomization (MR)-based claim of causality between elevated blood counts and JAK2 V617F clonality. I have described my concerns and a few other questions below.

1. The support for a causal link between higher platelet and monocyte counts and JAK2 V617F clones seems rather weak to me. The major question here is whether PLT- and MONO-increasing variants could directly promote clonal expansion of JAK2 V617F, which seems quite plausible -- in which case the key assumption of the IVW MR analysis would be violated. The three other MR methods produced non-significant or much less significant results, which the authors attributed to reduced power, but it is hard to really know from the data currently presented. The authors do have potential routes to increasing the power of their analysis, e.g., by using more powerful genetic instruments (from summary statistics of studies much larger than the N~30K INTERVAL participants currently used to estimate effect sizes) and/or by expanding to the full N=470K UK Biobank WES data set.

2. Were UK Biobank MPN cases restricted to participants who had existing diagnoses at the time of blood draw (i.e., initial assessment)? I could not seem to find this information in the Methods. If the ICD10 data included incident cases, this could be part of the reason that only ~10-17% of individuals had evidence of JAK2 V617F.

3. Related to the above observation of unusually low JAK2-V617F positivity, the authors "raise the possibility that many individuals in the UKBB-MPN cohort were those with elevated blood counts driven by high PGS" (p.6). This is a testable hypothesis: how do the distributions of the relevant PGSs among UKBB-MPN cases compare to controls?

4. The definitions of "odds ratios" (OR) reported throughout the manuscript need to be explained more clearly given that the independent variables are usually continuous (e.g., PGSs). Are the PGSs in units of standard deviations of the blood traits, and do the ORs indicate changes in odds per predicted 1 s.d. increase?

5. I found the abstract hard to understand on a first reading, in part because of the meaning of OR and also the mention of "an 'MPN' phenotype" without definition.

Author Rebuttal to Initial comments

Response to referees

Referee expertise:

Referee #1: Genetics, cancer, myeloproliferative

neoplasms Referee #2: Genetics, complex traits, clonal

hematopoiesis

Reviewers' Comments:

Re: We thank our Reviewers for their insightful comments and constructive suggestions for improving our manuscript. We have now addressed the questions point by point by introducing more powerful datasets, performing additional analyses and updating our manuscript/supplementary file accordingly. In this response letter, we copy the results/figures/tables in the old manuscript wherever relevant in purple, reply to each point (including the Extended Figure/Table that are only for this letter) in blue and update our manuscript with additional comments/figures/tables in red. For the ease of cross-reference for Reviewers, our responses may show duplicates of results/figures/tables for different questions.

Reviewer #1:

Remarks to the Author:

This paper by Guo and colleagues investigates the relationship between polygenic scores (PGS) for hematologic traits and clonal selection of JAK2V617F as well as subsequent development of myeloproliferative neoplasms (MPNs). PGS for monocyte count and plateletcrit were associated with JAK2 clonal expansion, while PGS for red cell traits were associated with PV and platelet trait PGS were associated with ET. Interestingly, extreme blood cell trait PGS may account for some false diagnoses of triple negative MPNs. The manuscript is well-written, employs appropriate methods and contributes novel etiologic understanding to MPN susceptibility. Below are comments for further consideration.

- Results (end of paragraph 1) mention studying “clonal dynamics”. As this study investigates JAK2-V617F at one timepoint, I suggest rephrasing as longitudinal data is not available on clonal trajectories.
 Re: We have removed ‘and its subsequent clonal dynamics’ and the revised sentence now reads “We wished to understand if germline polygenic loci that underlie blood cell traits influence the strength of clonal selection on JAK2V617F” (page 3).

- More details surrounding JAK2-V617F mutation detection in UK Biobank WES data would be helpful. What was the distribution for the sequencing coverage depth in this region? What caller(s) were used for calling JAK2-V617F? Were any exclusion criteria used to exclude samples with low coverage or remove reads with low base quality/mapping quality? As the frequency of JAK2-V617F was low in UK Biobank MPN cases, it is critical to detail the detection approach and outline limitations/thresholds of detection to put the resulting findings in context.

Re: We thank the Reviewer for their valuable advice. We agree it is important to consider why the rate of JAK2V617F positivity is low in UKBB and importantly, whether any sequencing metrics could be impacting on our findings or conclusions regarding JAK2 mutated versus unmutated patients.

The mpileup function of Samtools 1.9 was used with the FASTA file of the GRCh38 assembly and the parameter “-r chr9:5073767-5073775” to calculate the number of mutant reads and coverage of each base around the V617 hotspot. We excluded reads with base quality <13 using the base quality filter in the Samtools mpileup tool. We have now added additional details of JAK2V617F mutation detection in the Methods section (page 16): “The mpileup function of Samtools 1.9 was used with the FASTA file of

GRCh38 assembly and the parameter "-r chr9:5073767-5073775" to calculate the number of mutant reads and coverage of each base around the V617 hotspot. We excluded reads with base quality <13 using the base quality filter in Samtools mpileup tool. *JAK2V617F* clone size was measured by VAF in the UKBB-WES cohort. Descriptions of the 200k UKBB-WES can be found in Kar *et al*¹⁵.

The mean number of reads for *JAK2* V617F hotspot was 21.5 (median=21 and s.d.=6.4; **Figure S3e**). The mean coverage of all exons of *JAK2* was 27.5, (median=27 and s.d.=9.6; **Figure S3f**), which is the lowest among all CH genes (see Supplementary Table 1 in Kar *et al.*). This may result in decreased sensitivity of detection of small clones (e.g. VAF <0.1). We now clarify this at the beginning of the Results section (page 3) and add an additional figure (Figure S3) as follows: "We studied the germline characteristics of individuals in UK-Biobank (UKBB) with and without *JAK2V617F*. Of 162,534 genetically unrelated individuals with European ancestry within the UKBB whole exome sequencing cohort (200k UKBB-WES as our main discovery dataset; Methods), we identified 540 individuals with one or more mutant reads for *JAK2V617F* (0.3%, median variant allele frequency [VAF] 0.056, range 0.019-1, **Figure S2**; "UKBB-*JAK2V617F* cohort"). The lower rate of *JAK2V617F* in the UKBB-WES, compared to other population studies^{6,7}, could be explained by its low sequencing coverage (x21.5) as also reported previously¹⁵ (**Figure S3**)".

The analysis of the pileup output (including read quality for low-quality reads) showed that the distribution of quality scores in the 200k UKBB samples was broadly comparable between *JAK2V617F* positive, *JAK2V617F* negative and other *JAK2* loci (**Figure S3a, b and c**). Furthermore, we did not see a significant difference ($P_{\text{Difference}} = 0.53$) in the coverage between PV-*JAK2V617F* carriers and non-carriers (**Figure S3d**). Therefore, we are reassured that our clinical findings for *JAK2V617F* positive versus negative individuals were not due to differences in sequencing depth or quality of sequencing specifically affecting *JAK2V617F* negative individuals. These data have been added as a new Figure S3 together with the pile up data above and is shown below.

We also discuss other potential reasons in response to Question 2 from Reviewer #2 (Reviewer #2-Q2) regarding the low rate of *JAK2V617F* in UKBB-WES. Please also see this consolidated response below for Reviewer #2-Q2.

Supplementary Figure 3 Summary plot for *JAK2V617F* mutation detection in the 200k UK Biobank WES data. Shown in order are the percentages of read quality scores in *JAK2V617F*-positive (a), *JAK2V617F*-negative (b) and other reads (c) in the same exon in *JAK2*. Coverage distribution within PV between *JAK2V617F* carriers and non-carriers is shown in (d) and the distribution of coverage at the V617F hotspot (mean=21.5, median=21, standard deviation [s.d.]=6.4) and all exons of *JAK2* (mean=27.5, median=27, s.d.=9.5) are shown in (e) and (f) respectively, with median values indicated by dash lines. The analysis of the pileup output (including read quality for low-quality reads) showed that the distribution of quality scores in the 200k UKBB samples was broadly comparable between *JAK2V617F* positive, *JAK2V617F* negative and other *JAK2* loci (a, b and c). We did not see a significant difference ($P_{\text{Difference}} = 0.53$) in the coverage between PV-*JAK2V617F* carriers and non-carriers (d).

- It appears that MPN cases were included in the analyses on *JAK2*-V617F presence and clone size. Did the authors perform any stratified/sensitivity analyses among MPN-free individuals to ensure effects of PGSs on *JAK2* were not driven by the subset of MPN cases?

Re: We thank the reviewer for this important suggestion.

In the manuscript, we showed a negative association between PGS for mean reticulocyte volume (PGS_{MRV}) and immature reticulocyte fraction (PGS_{IRF}) ($P = 4.2 \times 10^{-4}$, 0.0018 respectively with FDR < 0.05) and small *JAK2V617F* clones. We also found significant positive associations with small *JAK2V617F* clones for the PGSs of plateletcrit, (PGS_{PCT}) and monocyte count (PGS_{MONO}) ($P = 9.5 \times 10^{-4}$ and 0.0036 respectively with FDR < 0.05). Germline predisposition to high MONO and PCT were positively associated with large *JAK2V617F* clones at modest significance ($P = 0.033$ and 0.0022 respectively; FDR = 0.31 and 0.064; **Figure**

1a).
a

b

Figure 1 Germline polygenic risk for haematological traits and association with *JAK2V617F* positivity (small and large clones). PGSs in panel a) are the identified significant associations for small clone size of *JAK2V617F* (FDR < 0.05) compared with the PGS of CH and MPN (Table S3). Panel b) presents the causal effects estimated by four MR methods for the exposure traits whose PGSs were found to have significant predisposition risks for *JAK2V617F* positivity (Table S4). The MR result of MRV was not available for the lack of corresponding GWAS summary data in INTERVAL.

Following the reviewer’s suggestion, we repeated the analysis in this revision excluding MPN cases. We found consistent significant effects on small *JAK2V617F* clones ($P < 0.05$; n cases for small clones decreased from 397 to 349; Table S4), while the effects on large *JAK2V617F* clones were attenuated but with consistent directions (n decreased from 143 to 49).

Supplementary Table 4 Germline polygenic associations with *JAK2V617F* positivity in the UKBB-*JAK2V617F* cohort excluding MPN cases.

PGS	Small clones (VAF < 0.1)			Large clones (VAF > 0.1)				
	OR	95% CI	P	OR	95% CI	P		
MRV	0.88	0.79	0.98	0.020	0.79	0.59	1.07	0.13
IRF	0.88	0.79	0.98	0.023	0.79	0.58	1.07	0.13
MONO	1.16	1.04	1.29	0.0080	1.16	0.86	1.56	0.32
PCT	1.16	1.04	1.30	0.0074	1.06	0.78	1.43	0.73
CH	1.05	0.95	1.18	0.34	1.11	0.83	1.49	0.49
MPN-46/1	1.31	1.18	1.44	1.7E-07	1.45	1.11	1.90	0.0069
MPN-other	1.38	1.25	1.53	6.6E-10	2.11	1.65	2.71	2.9E-09

We have now updated the Results section (page 4) as: Repeating the analysis above excluding MPN cases still demonstrated a significant association between PGS_{PCT}/PGS_{MONO} and small *JAK2V617F* clones ($P < 0.013$ Bonferroni corrected; n cases = 349 and 49 for small and large clones respectively; **Table S4**), suggesting that the effects of PGS on *JAK2V617F* are not driven by the subset of MPN cases.

- Some additional comment on the change in the magnitude of associations of the monocyte count and plateletcrit PGS by large vs. small clone size could be helpful, rather than focusing on significant vs. non-significant associations. Such dose response relationships are helpful for establishing causality.

Re: We thank the reviewer for this comment.

In the manuscript, we identified PCT and MONO as causal risk factors for the presence of a *JAK2V617F* clone based on 200k UKBB-WES data (n cases vs. controls = 540 vs. 161,994) using the MR-IVW method (OR_{IVW} = 1.5, 95% CI 1.2-1.8, $P = 4.1 \times 10^{-5}$ for PCT; OR_{IVW} = 1.3, 95% CI 1.1-1.6, $P = 0.002$ for MONO; Figure 1b). However, only MONO showed modest significance in MR-Egger (OR = 1.47 [95% CI = 1.04-2.09]; $P = 0.031$) with no significance for PCT (OR = 1.17 [95% CI = 0.79-1.75], $P = 0.435$).

Figure 1 Germline polygenic risk for haematological traits and association with *JAK2V617F* positivity (small and large clones). PGSs in panel a) are the identified significant associations for small clone size of *JAK2V617F* (FDR < 0.05) compared with the PGS of CH and MPN (Table S3). Panel b) presents the causal effects estimated by four MR methods for the exposure traits whose PGSs were found to have significant predisposition risks for *JAK2V617F* positivity (Table S4). The MR result of MRV was not available for the lack of corresponding GWAS summary data in INTERVAL.

Adopting the advice from Q1-Reviewer #2, we extended these analyses to the full UKBB-WES dataset made recently available. We repeated the calling of *JAK2V617F* mutations (1,125 in total) and reran all MR analyses based on the outcome-GWAS in this full data set (n = 1,125 cases vs. 338,919 controls). We

have added this as a new Figure S4, also shown below.

Supplementary Figure 4 Case-control GWAS of *JAK2V617F* positivity in unrelated individuals in the UKBB-WES cohort ($n = 1,125$ cases vs. 338,919 controls). The genetic relatedness matrix was built on the after-imputation QC-ed variants after LD pruning with an r^2 threshold of 0.2. Age, sex, WES batch and 10 PCs were fitted as covariates. The blue and red line represents the significance threshold of 10^{-6} and 5×10^{-8} .

By gaining the statistical power of a larger cohort, we found high consistency in the estimates of causal effects and a large improvement in the estimation accuracy across all four main MR methods for both MONO ($P < 0.05$ for simple median, IVW and Egger) and PCT ($P < 0.05$ for all four methods). We have updated **Figure 1b** with these new results, and added a new **Table S7** to the revised manuscript. This additional analysis across a larger UKBB-WES cohort ($n = 1,125$ cases and 338,919 controls vs. 540 cases and 161,994 controls) also validates our originally submitted findings. We update this in the Results section (page 4) that “Both PCT and MONO showed significant causality on the presence of a *JAK2V617F* clone based on the inverse-variance weighted MR method (IVW¹⁷), and demonstrated consistent effect estimates using two other MR methods (simple median and weighted median), suggesting that a higher monocyte count and a higher PCT cause a detectable *JAK2V617F* clone (**Table S6**). Extending this analysis to the full UKBB-WES ($n = 1,125$ *JAK2V617F*, 338,919 controls) validated these findings with greater estimation accuracy ($OR_{IVW} = 1.52$ [95% CI = 1.29-1.78] and $P = 3.0 \times 10^{-7}$ for PCT; $OR_{IVW} = 1.3$ [95% CI = 1.15-1.49] and $P = 4.6 \times 10^{-5}$ for MONO; **Figure 1b**; **Table S7**)”.

Figure 1 Germline polygenic risk for haematological traits and association with *JAK2V617F* positivity (small and large clones). PGSs in panel **a**) are the identified significant associations for small clone size of *JAK2V617F* (FDR < 0.05) compared with the PGS of CH and MPN (Table S3). Panel **b**) presents the causal effects estimated by four MR methods for the exposure traits whose PGSs were found to have significant predisposition risks for *JAK2V617F* positivity (Table S7). The MR result shown was based on the GWAS summary statistics in the full UKBB (Figure S4). Results based on the main discovery set (200k UKBB) are shown in Table S6. The MR result of MRV was not available for the lack of corresponding GWAS summary data in INTERVAL. MRV, mean reticulocyte volume; IRF, immature fraction of reticulocytes; MONO, monocyte count; PCT, plateletcrit; CH, clonal haematopoiesis; MPN, myeloproliferative neoplasms.

Supplementary Table 7 Forward causal inference of *JAK2V617F* positivity across four main methods based on the *JAK2V617F* GWAS in full UKBB (n = 1,125 cases vs. 338,919 controls

Outcome	Exposure	#Instruments	Methods	OR (95% CI)	P
JAK2V617F	IRF	373	Simple_median	0.87 (0.69-1.11)	0.26
			Weighted_median	0.78 (0.61-1.01)	0.058
			IVW	0.84 (0.71-1)	0.047
			Egger	0.87 (0.64-1.18)	0.36
	MONO	688	Simple_median	1.26 (1.03-1.54)	0.024
			Weighted_median	1.2 (0.96-1.51)	0.10
			IVW	1.31 (1.15-1.49)	4.6E-05
			Egger	1.33 (1.03-1.71)	0.026
	PCT	703	Simple_median	1.41 (1.16-1.71)	6.1E-04
			Weighted_median	1.4 (1.13-1.72)	1.7E-03
			IVW	1.52 (1.29-1.78)	3.0E-07
			Egger	1.56 (1.13-2.17)	7.5E-03

As suggested by the reviewer, we also assessed dose-response relationships, as opposed to just the significance of the causality. Below, we show the dose-response relationship between the genetic associations with the risk factor (i.e., MONO/PCT) and those with the outcome (i.e., JAK2V617F positivity), with the causal effect assessed by MR-Egger (OR = 1.33 [95% CI = 1.03-1.71] for MONO and 1.56 [95% CI = 1.13-2.17] for PCT). We see the strongest dose-response relationship for PCT, followed by MONO and IRF with the estimated regression coefficients of 0.45 (s.e. = 0.17), 0.28 (s.e. = 0.13) and -0.14 (s.e. = 0.16) respectively. We have added a plot for this dose-response relationship in **Figure S5**). Note that the genetic variants that were used as IVs have been oriented in the analyses so that all of the associations with exposure were positive.

Supplementary Figure 5 MR plots for the forward causality between a blood cell trait and JAK2V617F

positivity. IVs for an exposure (e.g., IRF) were selected from the published GWAS in full UKBB with effect sizes estimated in INTERVAL (association $P < 5e-8$ in UKBB and also < 0.05 in INTERVAL). The effect sizes for outcome (*JAK2V617F* positivity) were based on the corresponding GWAS in full UKBB ($n = 1,125$ cases vs. 338,919 controls). The strongest dose-response relationship is for PCT, then for MONO and IRF, with the estimated Egger regression coefficients of 0.45 (s.e. = 0.17), 0.28 (s.e. = 0.13) and -0.14 (s.e. = 0.16) respectively.

We also formally tested if these causal estimates may have been biased by the presence of any outlier variants exerting undue influence, and whether there was any such significant distortion in the causal estimate before and after removing outliers (if any) using MR pleiotropy residual sum and outlier (MR-PRESSO) analyses. We found 1 outlier variant for MONO and 9 outliers for PCT, but the distortion test before and after removing outliers was not significant for either MONO or PCT ($P_{\text{Distortion}} = 0.91$ and 0.48 respectively). Similarly, the result of penalized MR-Egger, which downweights the contribution of genetic variants with outlying ratio estimates, also suggested that the significance of the identified causal effect was not strongly affected by outlier variants (OR = 1.24 [95% CI = 0.98-1.59; $P = 0.077$] for MONO and 1.53 [95% CI = 1.17-2.00; $P = 0.0020$] for PCT).

In the manuscript, we keep the association results based on the main discovery dataset (the 200k UKBB-WES) as Figure 1a, and add the MR results based on the full UKBB-WES as **Figure 1b**. Here are the changes in the main texts (page 4):

To understand the causal relationship between these associations we undertook two-sample Mendelian randomisation analysis. This approach requires GWAS estimates for the exposure (blood traits) and the outcome (*JAK2V617F* positivity; **Figure S4**) from two independent sources. Hence, we used genetic instruments for the blood cell traits identified from UKBB but with effect size estimates (and standard errors [s.e.]) from INTERVAL¹⁷ ($n=30,305$), an external independent cohort (MRV was excluded due to lack of data in INTERVAL, see **Methods**). Both PCT and MONO showed significant causality on the presence of a *JAK2V617F* clone based on the inverse-variance weighted MR analysis (IVW¹⁸), and demonstrated consistent effect estimates using two other MR methods (simple median and weighted median), suggesting that a higher MONO and a higher PCT cause a detectable *JAK2V617F* clone (**Table S6**). Extending this analysis to the full UKBB-WES ($n = 1,125$ *JAK2V617F*, 338,919 controls) validated these findings with greater estimation accuracy (OR_{IVW} = 1.52 [95% CI = 1.29-1.78] and $P = 3.0 \times 10^{-7}$ for PCT; OR_{IVW} = 1.3 [95% CI = 1.15-1.49] and $P = 4.6 \times 10^{-5}$ for MONO; **Figure 1b**; **Table S7**). IVW assumes that the germline loci that drive MONO and PCT have no direct causal effect on driving a *JAK2V617F* clone (i.e., there are no direct causal effects of the genetic instruments on the outcome). We found no evidence of pleiotropy using the MR-Egger¹⁹ test with the estimated intercept terms not significantly different from zero ($P = 0.84$ and 0.90 based on the full UKBB cohort). The causal relationship was also significant for PCT and MONO (OR_{Egger} = 1.56 [95% CI=1.13-2.17] and $P = 0.0075$ for PCT; OR_{Egger} = 1.33 [95% CI=1.03-1.71] and $P = 0.026$ for MONO; **Table S7**; **Figure S5**). We also demonstrated that these causal estimates were not biased by any potential pleiotropic outlier variants and were highly consistent with the outlier-corrected causal estimates (OR_{Outlier-corrected} = 1.46 [95% CI = 1.27-1.67] and $P = 7.2 \times 10^{-8}$ for PCT; OR_{Outlier-corrected} = 1.30 [95% CI = 1.14-1.48] and $P = 6.2 \times 10^{-5}$ for MONO; **Methods**).

In the next comment about reverse causality (*JAK2V617F* -> MONO/PCT) the reviewer pointed out a potential weak genetic instrument bias. Indeed, MR-Egger regression assumes the SNP-exposure associations to be known rather than estimated, an assumption called NO Measurement Error (NOME). Such an assumption can lead to a dilution of the causal estimate. We therefore examined if the estimated causal effect was diluted for both forward and reverse causality. (Please see the corresponding analysis for reverse causality below.) In our analysis, the estimates of r^2 were smaller than 90% for both MONO and PCT, indicating a violation of NOME for MR-Egger. We therefore applied a correction to the Egger estimates to counteract the dilution using an established method of simulation extrapolation (SIMEX)²⁻⁴. The corrected causal effect estimates are 0.59 (s.e. = 0.21, $P = 0.0058$) for PCT and 0.36 (s.e. = 0.16; $P = 0.023$) for MONO; the corresponding OR values are 1.80 (95% CI = 1.20-2.72) and 1.43 (95% CI = 1.05-1.96) (Rebuttal Fig. 1b and c).

Rebuttal Figure 1 Simulation extrapolation (SIMEX) applied to the MR-Egger estimates for the forward causality data. The large blue dot represents the Egger estimate. Each small blue dot indicates a new value of Egger estimate averaged across many repeats of simulation for a same value of λ . The average value decreases as the magnitude of λ increases and the regression dilution effect gets stronger. + shows the adjusted estimate predicted by the model at the value $\lambda = -1$, which conceptually represents a setting with no measurement error (i.e., NOME satisfied).

- "Importantly we did not detect significant reverse causality from *JAK2-V617F* positivity to either PCT or MONO" – I commend the authors for performing the two-sample MR to assess directionality of effect. That said, I am curious what proportion of variance of the blood cell traits as well as *JAK2-V617F* was explained by the instruments used? Could a poor instrument for *JAK2-V617F* result in limited ability to identify reverse causality? Additionally, was *JAK2 46/1* haplotype information utilized in the instrument?

Re: We thank the reviewer for this comment!

We agree that poor instruments for *JAK2V617F* could result in limited ability to identify causality. In the analysis for reverse causality (*JAK2V617F* -> MONO/PCT), we selected 21 independent genetic IVs (association $P < 1e-5$) based on a GWAS summary data for *JAK2V617F* positivity ($n = 540$ cases vs. 161,994

controls). As expected, the I^2 values were low (34.5% for MONO and 34.4% PCT respectively) because of the use of a suggestive significance cut-off (association $P < 1e-5$). The estimated proportion of variance of *JAK2V617F* explained by the 21 IVs was 20%, versus 0.09% for MONO and 0.19% for PCT.

To get stronger instruments, we now have rerun the MR analysis for reverse causality based on the GWAS for *JAK2V617F* positivity in the full UKBB ($n = 1,125$ cases vs. 338,919 controls). We selected 11 independent IVs ($LD r^2 < 0.05$) under an association P cut-off stronger than the previous one ($P < 1e-6$) to maximize the number of IVs (only 4 in total under $P < 5e-8$) and also the strength of IVs in MR-Egger (IVs with F-statistic > 10 ; $I^2 = \sim 91\%$). The results did not show significant dose-response relationship either for MONO ($OR_{Egger} = 0.994$; 95%CI = 0.96-1.03; $P = 0.74$) or for PCT ($OR_{Egger} = 0.93$; 95%CI = 0.86-1.01; $P = 0.075$;

Table S9; Figure S6). The estimated proportion of variance explained by the 11 IVs were 9.4% for *JAK2V617F* positivity, 0.063% for MONO and 0.27% for PCT. (The results based on the IVs with $P < 1e-5$ were similar.)

Supplementary Figure 6 MR plots for the reverse causality between *JAK2V617F* positivity and a blood cell trait based on the independent IVs ($LD r^2 < 0.05$) selected from the GWAS for *JAK2V617F* positivity in full UKBB ($n = 1,125$ cases vs. 338,919 controls) under association $P < 1e-6$.

Supplementary Table 9 Reverse causal inference of *JAK2V617F* positivity and haematological traits across four main methods (#instruments = 11 with association $P < 1e-6$; $LD r^2 < 0.05$) based on the *JAK2V617F*-positivity GWAS in full UKBB ($n = 1,125$ cases vs. 338,919 controls).

Exposure	Outcome	Methods	beta (s.e.)	P
		Simple median	-0.015 (0.012)	0.23

JAK2V617F	IRF	Weighted median	-0.031 (0.01)	0.0019
		IVW	-0.027 (0.011)	0.010
		Egger (intercept)	-0.046 (0.026) 0.009 (0.012)	0.082 0.44
	MONO	Simple median	0.029 (0.012)	0.014
		Weighted median	0.026 (0.01)	0.010
		IVW	0.025 (0.008)	0.0018
		Egger (intercept)	-0.006 (0.017) 0.015 (0.008)	0.74 0.049
	PCT	Simple median	0.079 (0.014)	5.14E-08
		Weighted median	0.049 (0.015)	0.0011
		IVW	0.031 (0.022)	0.16
		Egger	-0.072 (0.041)	0.075
		(intercept)	0.052 (0.019)	0.0054

However, the results seem to suggest a modest significance for the reverse causality for IRF ($P_{\text{Egger}} = 0.082$ and $P_{\text{Egger-intercept}} = 0.44$)⁵. The estimated proportion of variance explained by the 11 IVs was 0.10% for IRF. Despite the above analyses, we cannot rule out the possibility of a causal relationship from JAK2 mutation to haematological traits (MONO/PCT) if more powerful GWAS data for JAK2 clonal expansion become available in the future.

We comment on the results of reverse causality In the end of the MR section (page 5) that “Importantly, any reverse causal effect we detected for MONO and PCT was subtle and with pleiotropic effects ($P_{\text{Egger}} > 0.05$ and $P_{\text{Egger-intercept}} < 0.05$; Table S9; Figure S6)”.

We confirm that the JAK2 46/1 haplotype information was utilised in the IVs. The IVs in the analysis were selected to be independent of each other with LD $r^2 < 0.05$. One of the IVs (rs2225125; 9:4998639 A>G, GRCh37), which is the top signal of the GWAS for JAK2V617F positivity in the full UKBB (Extended Table 1), is in strong LD with the four main tag SNPs of JAK2 46/1 haplotype (LD $r^2 > 0.94$).

Extended Table 1 Results of the top association SNP (rs2225125) and the four main tag SNPs of JAK2 46/1 haplotype in the GWAS for JAK2V617F positivity in full UKBB.

SNP	chr_bp	A1	A2	A1_freq	b	se	p
rs2225125	9_4998639	G	A	0.26	0.72	0.05	3.06E-41
rs12343867	9_5074189	C	T	0.26	0.70	0.05	4.40E-40
rs3780367	9_5068755	G	T	0.26	0.70	0.05	1.11E-39
rs10974944	9_5070831	G	C	0.26	0.68	0.05	3.42E-37

rs1159782 9_5078117 C T 0.26 0.69 0.05 5.29E-39

- It does not appear the authors investigated how actual phenotypic measures of blood cell traits are related to JAK2-V617F positivity and clone size. Such analyses would help support the PGS findings and would provide further evidence for the “alternative” explanation for the PGS associations presented in Discussion paragraph 2. This should be possible as phenotypic measurements of blood cell counts are available in UK Biobank.

Re: We thank the reviewer for the comment.

In the paper, we identified four significant germline effects (FDR < 0.05) on small clones of JAK2V617F positivity (Figure 1a).

Figure 1 Germline polygenic risk for haematological traits and association with JAK2V617F positivity (small and large clones). PGSs in panel a) are the identified significant associations for small clone size of JAK2V617F (FDR < 0.05) compared with the PGS of CH and MPN (Table S3). Panel b) presents the causal effects estimated by four MR methods for the exposure traits whose PGSs were found to have significant predisposition risks for JAK2V617F positivity (Table S4). The MR result of MRV was not available for the lack of corresponding GWAS summary data in INTERVAL.

Following the reviewer’s comment, we repeated the association analysis replacing PGSs by the corresponding phenotypic measures conditioning on the same covariates including PGSs of CHIP and MPN,

age, sex and principal components controlling for population stratification. Except for PCT, the estimated phenotypic effects were around zero for IRF, MRV and MONO ($P > 0.05$; **Rebuttal Figure 2**). This is not surprising to us because the association of the PGS of a trait X with a trait Y does not necessarily mean the phenotype of X is also associated with Y. We derived the relationship among them as below.

Rebuttal Figure 2 Germline and phenotypic associations of haematological traits with *JAK2V617F* positivity (small and large clones) in the main discovery dataset (200k UKBB-WES; a and b) and full UKBB-WES (c and d) respectively. The shown PGSs are the identified significant associations for small clone size of *JAK2V617F* (FDR < 0.05) compared with the PGS of CH and MPN.

In quantitative genetics, the observed phenotypic (P) value of a trait can be expressed as the average phenotype carrying specific genotypes (G) plus a deviation from the genotypic value contributed by environmental (E) factors, i.e., $P = G + E$. Through mathematical derivation, we show that the correlation of genetic values of an exposure X (G_X) with an outcome Y (P_Y) is a function of the correlation between the phenotype of the outcome (P_Y) and environmental deviation (E), assuming no correlation between G_X and E (i.e., $cov(G_X, E) = 0$):

$$\begin{aligned}
 \text{cor}(P_Y, P_X) &= \frac{\text{cov}(P_Y, G_X + E)}{\text{sd}(P_Y) * \text{sd}(G_X + E)} = \frac{\text{cov}(P_Y, G_X) + \text{cov}(P_Y, E)}{\sqrt{\text{var}(P_Y) * (\text{var}(G_X) + \text{var}(E))}} \\
 &= \frac{\text{cov}(P_Y, G_X)}{\sqrt{\text{var}(P_Y) * \text{var}(G_X) + \text{var}(P_Y) * \text{var}(E)}} + \frac{\text{cov}(P_Y, E)}{\sqrt{\text{var}(P_Y) * \text{var}(G_X) + \text{var}(P_Y) * \text{var}(E)}} \\
 &= \frac{\frac{\text{cov}(P_Y, G_X)}{\text{sd}(P_Y)} * \text{sd}(G_X)}{\sqrt{\text{var}(P_Y) * \text{var}(G_X) + \text{var}(P_Y) * \text{var}(E)}} + \frac{\frac{\text{cov}(P_Y, E)}{\text{sd}(P_Y) * \text{sd}(E)}}{\sqrt{\text{var}(P_Y) * \text{var}(G_X) + \text{var}(P_Y) * \text{var}(E)}} \\
 &= \frac{\text{cor}(P_Y, G_X)}{\sqrt{1 + \frac{\text{var}(E)}{\text{var}(G_X)}}} + \frac{\text{cor}(P_Y, E)}{\sqrt{\frac{\text{var}(G_X)}{\text{var}(E)} + 1}} \tag{1}
 \end{aligned}$$

Since

$$h_x^2 = \frac{\text{var}(G_X)}{\text{var}(G_X) + \text{var}(E)}$$

$$\frac{1}{h_x^2} = \frac{\text{var}(G_X) + \text{var}(E)}{\text{var}(G_X)}$$

$$\frac{1}{h_x^2} - 1 = \frac{\text{var}(E)}{\text{var}(G_X)}$$

$$\frac{\text{var}(E)}{\text{var}(G_X)} = \frac{1}{h_x^2} - 1$$

$$\frac{\text{var}(G_X)}{\text{var}(E)} = \frac{h_x^2}{1 - h_x^2}$$

Then Equation (1) can be written as

$$\begin{aligned}
 \text{cor}(P_Y, P_X) &= \frac{\text{cor}(P_Y, G_X)}{\sqrt{1 + \frac{\text{var}(E)}{\text{var}(G_X)}}} + \frac{\text{cor}(P_Y, E)}{\sqrt{\frac{\text{var}(G_X)}{\text{var}(E)} + 1}} \\
 &= \frac{\text{cor}(P_Y, G_X)}{\sqrt{\frac{1}{h_x^2}}} + \frac{\text{cor}(P_Y, E)}{\sqrt{\frac{1}{1 - h_x^2}}} \\
 &= \text{cor}(P_Y, G_X) * \sqrt{h_x^2} + \text{cor}(P_Y, E) * \sqrt{1 - h_x^2}. \tag{2}
 \end{aligned}$$

For an exposure, it is possible that environmental effects counteract genetic effects, making the P_Y (that

are finally expressed at an observable level) negatively correlated with E, resulting in the phenotypic effects on the outcome attenuating or acting in an opposite direction (Equation (2)).

Actually, this is also one of the advantages of using genetic variants as proxies of an exposure in MR, where genetic variants can indicate long-term levels of exposure while phenotypic measurements could suffer from measurement error and/or high levels of variability due to unknown/unmeasured environmental factors⁶. However, we also caution that the observed germline associations (and causal associations) with *JAK2V617F* positivity should be replicated in an independent dataset.

We add this point to the Discussion section (page 14):

We note that the causal relationship inferred by MR does not necessarily suggest a direct association between baseline MONO/PCT levels themselves on positivity for *JAK2V617F*, as blood cell parameters can suffer from measurement error and/or high levels of variability due to unknown/unmeasured environmental factors. These factors can counteract genetic effects, the latter providing better proxies for stable long-term exposure of a trait on an outcome.

- The Figure 1 legend is missing information on abbreviations used in the figure. Re: Have added.

- For the analyses of PGS impact on MPN subtype at diagnosis, myelofibrosis results are missing. I suspect this was a sample size issue and therefore myelofibrosis analyses were not conducted, but if myelofibrosis analyses were performed and the results were null this should be stated.

Re: In the manuscript, we stated the result in the somatic driver mutation section that

“We did not see any significant associations between PGS for blood cell traits and progression of disease to MF that replicated in both cohorts, consistent with the notion that disease transformation to MF is driven more by acquisition of additional somatic driver mutations”.

In the revised version, we also highlighted it in the Result section *Germline polygenic impact on MPN disease subtype at diagnosis* that “We also explored germline associations with presentation with primary MF and subsequent MF transformation. However, we did not see any significant associations that were replicated in the two cohorts” (page 9).

- Also for the MPN subtype associations, it was nice to see replication in UK Biobank MPN cases; however, a comparison of effect estimates between the two cohorts would be interesting, especially since UK Biobank had a much lower frequency of *JAK2-V617F* mutations among MPN cases.

Re: We thank the reviewer for the suggestion. In the supplementary file, we listed all the OR estimates across 29 traits in Figure S8a for MPN-patient cohort and S8b for UKBB-WES, with corresponding values in Table S10 and S11.

Following the reviewer’s suggestion, we formally tested the difference in the estimated regression coefficients of PGS between the MPN-patient cohort and the UKBB-MPN cohort for ET and PV respectively. Although only five (out of twenty-nine) germline predisposition factors (PGS_{PLT} and PGS_{PCT} for ET; PGS_{HGB}, PGS_{HCT} and PGS_{RBC} for PV) passed our significance threshold (FDR <0.05 in one cohort

and $P < 0.05$ in another; Figure S8 and Table S10 and S11), we only found subtle significant differences in the PV-associated effects for PGS_{PLT} ($\beta = 0.13$ [s.e. = 0.098; $P = 0.20$] vs. -0.13 [s.e. = 0.084; $P = 0.13$]; $P_{\text{difference}} = 0.05$) and PGS_{MONO} ($\beta = 0.22$ [s.e. = 0.097; $P = 0.026$] vs. -0.044 [s.e. = 0.082; $P = 0.59$]; $P_{\text{difference}} = 0.041$), suggesting the estimates of polygenic effects associated with ET/PV were overall consistent between the two cohorts. This reassured us that the polygenic predisposition effects on MPN appeared independent of the level of positivity of *JAK2V617F* due to sequencing differences between the cohorts.

- It was not clear in the somatic driver mutation section how *JAK2-V617F* carriers with chr 9 alterations were included in the analysis. I suspect a sizeable fraction of cases included both point mutations and chromosomal alterations.

Re: We apologise for not making this clear. In the somatic driver mutation section, we subdivided patients into three groups for ET after stratification: (1) *JAK2* heterozygous (*JAK2*_{het}, i.e., *JAK2* mutation without Chr9 alterations; $n = 308$), (2) *CALR*+ or *MPL*+ ($n = 169$), and (3) those negative for *JAK2*, *MPL* or *CALR* (triple-negative [TN]; $n = 81$). We did not perform a separate analysis for the *JAK2* homozygous group (*JAK2*_{hom}, i.e., *JAK2V617F* carriers with Chr9 aberrations) for ET because of limited sample size ($n = 16$; Figure 4a red panels). Indeed, homozygosity for mutant *JAK2* is less frequent and present at lower VAF in ET compared to PV.

For PV, 112 cases included 62 *JAK2*_{hom} and 50 *JAK2*_{het} (Figure 4b red panels). The differentiation pattern was not significantly different between PV-*JAK2*_{hom} and PV-*JAK2*_{het}.

Figure 4 Germline polygenic risks on ET and PV in the context of somatic mutations. ORs of the top and bottom PGS quintiles relative to that of the middle three quintiles in the MPN-patient cohort (red and

yellow) and the UKBB-MPN cohort (blue and grey) for ET (panel a) and PV (panel b) respectively. Mutation-stratified groups are shown wherever the data are available. Shown n is the sample size of ET (or PV) cases in a specific group. It is of note that *JAK2*-negative PV should be a rare diagnosis in PV (normally ~5%) but the number of such patients called in the UKBB-WES data appeared to be high (n = 121 out of 161; **Table S1**). It may be that many of these individuals do not have an underlying clonal disorder such as PV but present with secondary erythrocytosis e.g. due to smoking, alcohol, lung disease, where a germline predisposition to high HGB/HCT results in blood counts that mimic PV. However, there was limited statistical power for demonstrating any significance for the *JAK2*V617F PV subgroup, given the small sample size (UKBB-MPN cohort, n = 40). Furthermore, we cannot exclude the possibility that a mutant *JAK2* clone was missed during sequencing in some of these individuals.

In the Methods section, we clarified that "The stratification resulted in four groups for ET and three groups for PV, which were *JAK2*_{het} group (*JAK2*+C9-MPL-CALR-; n = 308), CALR+/MPL+ group (*JAK2*-C9-MPL+CALR-, *JAK2*-C9-MPL-CALR+ and *JAK2*-C9-MPL+CALR+; n = 169) and triple negative group (*JAK2*-C9-MPL-CALR-; n = 81) in MPN-patient cohort and one ET-all group in UKBB-MPN cohort as a replicate; PV-all (i.e., *JAK2*+ including 50 *JAK2*_{het} and 62 *JAK2*_{hom}) in MPN-patient cohort, *JAK2*+ PV and *JAK2*- PV in UKBB-MPN cohort" (page 21).

Reviewer #2:

Remarks to the Author:

This manuscript explores connections between blood cell trait polygenic scores (PGS) and myeloproliferative neoplasm (MPN)-related phenotypes. Specifically, the authors consider *JAK2* V617F clones in UK Biobank, MPN patients from a separate cohort, and MPN cases reported in UK Biobank. They identify associations between PGS and all of these traits, leading them to draw two main conclusions: (i) higher platelet and monocyte counts causally drive *JAK2* V617F clonality; and (ii) polygenic predisposition to blood cell traits influences risk of MPNs.

The latter analyses of MPNs constitute the bulk of the paper and seem generally thorough and well-supported, with several different analyses exploring variance explained by PGSs in healthy individuals versus MPN cases, independence of PGSs from known MPN risk variants, and robustness of effects across MPN somatic driver mutation classes and across cohorts. The idea that common-variant predisposition to higher or lower blood cell indices can substantially influence risk and/or clinical classification of MPNs (for which somatic mutations are usually the focus) seems quite interesting to me -- though I should caveat that I am not a hematologist and may not be up to date with literature in this field. I did find one other recent report of blood PGSs associated with acute lymphoblastic leukemia (PMID 34469753), and I noticed that Bao et al. (ref. 9) did observe genetic correlations between MPN risk and blood traits, so this idea is not entirely unprecedented.

I was less convinced by the Mendelian randomization (MR)-based claim of causality between elevated

blood counts and JAK2 V617F clonality. I have described my concerns and a few other questions below.

1. The support for a causal link between higher platelet and monocyte counts and JAK2 V617F clones seems rather weak to me. The major question here is whether PLT- and MONO-increasing variants could directly promote clonal expansion of JAK2 V617F, which seems quite plausible -- in which case the key assumption of the IVW MR analysis would be violated. The three other MR methods produced non-significant or much less significant results, which the authors attributed to reduced power, but it is hard to really know from the data currently presented. The authors do have potential routes to increasing the power of their analysis, e.g., by using more powerful genetic instruments (from summary statistics of studies much larger than the N~30K INTERVAL participants currently used to estimate effect sizes) and/or by expanding to the full N=470K UK Biobank WES data set.

Re: We thank the reviewer's suggestion about the plausible violation of a key instrument assumption in MR and the power issue for the forward causality (PCT/MONO -> JAK2V617F positivity). Our response below consolidates our response to a few relevant questions from Reviewer #1 as well.

The concern here is regarding a key assumption of instruments in MR, which is that instruments should associate with the outcome only through the exposure, a violation of which indicates directional pleiotropy. Satisfying this assumption implies that intervention on IVs should not result in any changes in outcome if the exposure remains constant⁷ [cite]. In the revision, we managed to justify this by carrying out additional sanity checks including estimating causal effects and any potential pleiotropic effects using

more powerful genetic instruments, and repeating the MR analyses for each exposure (IRF/PCT/MONO) excluding any instruments that could confound with outcome, i.e., those either associated with JAK2V617F positivity (association $P < 1e-6$), correlated with (LD $r^2 > 0.01$), or were near (within +/-5Mb region) those JAK2V617F-associated variants.

We thank the reviewer's advice specifically regarding the power of genetic instruments. Since the instruments associated with the exposure were already selected from the blood cells GWAS in the full UKBB (as far as we know the largest blood cells GWAS to date), and using genetic instruments with effect sizes estimated in UKBB were actually avoided due to the possibility of falling foul of the winner's curse, we adopt the Reviewer's second suggestion and have expanded the GWAS for the outcome (i.e., JAK2V617F positivity) to the full UKBB-WES data of 470K participants, in order to improve estimation accuracy of the genetic effects of instruments.

In the full UKBB data, we called 1,125 cases for JAK2V617F mutations and performed a case-control GWAS along with 338,919 controls (**Figure S4**).

Supplementary Figure 4 Case-control GWAS of the *JAK2V617F* positivity in the unrelated individuals in the UKBB-WES (n = 1,125 cases vs. 338,919 controls). The genetics relatedness matrix was built on the after-imputation QC-ed variants after LD pruning with r^2 threshold of 0.2. Age, sex, WES batch and 10 PCs were fitted as covariates. The blue and red line represents the significance threshold of 10^{-6} and 5×10^{-8} .

For comparison with the original results, we selected independent IVs ($LD\ r^2 < 0.05$) for exposure that were discovered at genome-wide significance threshold in the original blood cell GWAS ($P < 5e-8$ in UKBB) and also replicated in INTERVAL with modest significance ($P < 0.05$). We then reran the MR analysis based on this more powerful GWAS for *JAK2V617F* positivity. We do now see improved estimation accuracy with stronger significance for both MONO ($P < 0.05$ for simple median, IVW and Egger) and PCT ($P < 0.05$ across all four methods) (**Figure 1b**; **Table S7**). Considering the potential weakness of the evidence presented in the original results on the smaller UKBB-WES dataset, we have updated the main figure using these new results (**Figure 1b**), keeping the old analysis from the smaller UKBB-WES cohort analysis in the supplementary file (**Table S6**).

Figure 1 Germline polygenic risk for haematological traits and association with *JAK2V617F* positivity (small and large clones). PGSs in panel a) are the identified significant associations for small clone size of *JAK2V617F* (FDR < 0.05) compared with the PGS of CH and MPN (Table S3). Panel b) presents the causal effects estimated by four MR methods for the exposure traits whose PGSs were found to have significant predisposition risks for *JAK2V617F* positivity (Table S7). The MR result shown was based on the GWAS summary statistics in the full UKBB. Results based on the main discovery set (200k UKBB) are shown in Table S6. The MR result of MRV was not available for the lack of corresponding GWAS summary data in INTERVAL. MRV, mean reticulocyte volume; IRF, immature fraction of reticulocytes; MONO, monocyte count; PCT, plateletcrit; CH, clonal haematopoiesis; MPN, myeloproliferative neoplasms.

Supplementary Table 7 Forward causal inference of *JAK2V617F* positivity across four main methods based on the *JAK2V617F* GWAS in full UKBB (n = 1,125 cases vs. 338,919 controls).

Outcome	Exposure	#Instruments	Methods	OR (95% CI)	P
JAK2V617F	IRF	373	Simple_median	0.87 (0.69-1.11)	0.26
			Weighted_median	0.78 (0.61-1.01)	0.058
			IVW	0.84 (0.71-1)	0.047
	MONO	688	Egger	0.87 (0.64-1.18)	0.36
			Simple_median	1.26 (1.03-1.54)	0.024
			Weighted_median	1.2 (0.96-1.51)	0.10
			IVW	1.31 (1.15-1.49)	4.6E-05
	PCT	703	Egger	1.33 (1.03-1.71)	0.026
			Simple_median	1.41 (1.16-1.71)	6.1E-04
			Weighted_median	1.4 (1.13-1.72)	1.7E-03
			IVW	1.52 (1.29-1.78)	3.0E-07
			Egger	1.56 (1.13-2.17)	7.5E-03

As suggested by the Reviewer #1, we also assessed dose-response relationships, as opposed to just the significance of the causality. Below, we show the dose-response relationship between the genetic associations with the risk factor (i.e., MONO/PCT) and those with the outcome (i.e., JAK2V617F positivity) with the causal effect assessed by MR-Egger (OR = 1.33 [95% CI = 1.03-1.71] for MONO and 1.56 [95% CI = 1.13-2.17] for PCT). We see the strongest dose-response relationship for PCT, followed by MONO and IRF, with estimated regression coefficients of 0.45 (s.e. = 0.17), 0.28 (s.e. = 0.13) and -0.14 (s.e. = 0.16) respectively. We have added a plot for this dose-response relationship in **Figure S5**). Note that the genetic variants that were used as IVs have been oriented in the analyses so that all of the associations with exposure are positive.

Supplementary Figure 5 MR plots for the forward causality between a blood cell trait and *JAK2V617F* positivity. IVs for an exposure (e.g., IRF) were selected from the published GWAS in full UKBB with effect sizes estimated in INTERVAL (association $P < 5 \times 10^{-8}$ in UKBB and also < 0.05 in INTERVAL). The effect sizes for outcome (*JAK2V617F* positivity) were based on the corresponding GWAS in full UKBB ($n = 1,125$ cases vs. 338,919 controls). The strongest dose-response relationship is for PCT, then for MONO and IRF, with the estimated Egger regression coefficients of 0.45 (s.e. = 0.17), 0.28 (s.e. = 0.13) and -0.14 (s.e. = 0.16) respectively.

We also formally tested if these causal estimates may have been biased by the presence of any outlier variants exerting undue influence, and whether there was any such significant distortion in the causal estimate before and after removing outliers (if any) using MR pleiotropy residual sum and outlier (MR-PRESSO) analyses. We found 1 outlier variant for MONO and 9 outliers for PCT, but the distortion test before and after removing outliers was not significant for either MONO or PCT ($P_{\text{Distortion}} = 0.91$ and 0.48 respectively). Similarly, the result of penalized MR-Egger, which downweights the contribution of genetic variants with outlying ratio estimates, also suggested that the significance of the identified causal effect was not strongly affected by outlier variants (OR = 1.24 [95% CI = 0.98-1.59; $P = 0.077$] for MONO and 1.53 [95% CI = 1.17-2.00; $P = 0.0020$] for PCT).

Finally, to examine if any exposure trait-increasing instruments potentially promote *JAK2V617F* positivity, we extracted any PLT/MONO-IVs that were either associated *JAK2V617F* positivity at $P < 1 \times 10^{-6}$, in LD with, or within a +/-5Mb region of those associated variants. We then repeated the MR analysis excluding these extracted IVs. We did not see any major changes in the results (**Table S8**). We have updated this in the Results section (page 5):

Furthermore, to ensure the results were not confounded by the possibility that genetic loci used as instruments for the MR directly promote the outcome (i.e., *JAK2V617F* positivity), we repeated the analysis excluding genetic instruments that associated with *JAK2V617F* positivity (at association $P < 10^{-6}$), as well as those that correlated with (linkage disequilibrium [LD] $r^2 > 0.01$) or were in proximity to *JAK2V617F* variants (within the +/-5Mb region of those variants) and found no major changes (**Table S8**).

Supplementary Table 8 Forward causal inference of *JAK2V617F* positivity across four main methods with *JAK2V617F* GWAS in full UKBB ($n = 1,125$ vs. 338,919) excluding instruments that are either associated with *JAK2V617F* positivity at $P < 1 \times 10^{-6}$, in LD ($r^2 > 0.01$) with or within +/-5Mb region of those variants.

Outcome	Exposure	#Instruments	Methods	OR (95% CI)	P
JAK2V617F	IRF	319	Simple_median	0.9 (0.69-1.16)	0.40
			Weighted_median	0.76 (0.58-0.99)	0.042
			IVW	0.85 (0.71-1)	0.056
			Egger	0.81 (0.6-1.1)	0.17
	MONO	571	Simple_median	1.2 (0.96-1.49)	0.12
			Weighted_median	1.16 (0.91-1.46)	0.22
			IVW	1.28 (1.11-1.48)	7.19E-04
			Egger	1.31 (0.99-1.73)	0.061
	PCT	547	Simple_median	1.44 (1.15-1.81)	0.0014
			Weighted_median	1.53 (1.2-1.95)	5.24E-04
			IVW	1.61 (1.38-1.88)	1.10E-09
			Egger	1.97 (1.42-2.74)	4.59E-05

Lastly, we endeavoured to estimate the genetic correlation between PLT/MONO and JAK2V617F positivity to quantify to what extent the underlying genetic component was shared, and we have run an additional analysis to distinguish such genetic correlation from causation. However, no significant genetic correlation was found with JAK2V617F positivity either for PLT or MONO, which probably can be explained by the limited power in the GWAS data for JAK2V617F positivity (observed scale heritability = 0.0042 [s.e. = 0.0032; $P = 0.20$]). We reported this in the revised manuscript and caution that future studies with more powerful data may be helpful.

We summarised the discussion above in the Discussion section (page 13):

We identified that high PGS_{MONO} and high PGS_{PCT} were associated with detectable clone positivity for JAK2V617F. Our forward MR analyses suggest that the mode of action is via a route where the elevated blood counts themselves positively select on mutant JAK2, with evidence that an individual's genetically predicted MONO/PCT levels causally select on mutant JAK2. Indeed, clonal expansion of nascent mutated HSCs in MPN and CH has been suggested to be influenced by inflammation³⁷⁻⁴⁰. However, we caution that a sensitivity analysis distinguishing causality from potential genetic correlation should be considered in future studies with empowered GWAS data for JAK2 clone expansion⁴¹ (although no significant genetic correlation was detected in our data). We note that the causal relationship inferred by MR does not necessarily suggest a direct association between baseline MONO/PCT levels themselves on positivity for JAK2V617F, as blood cell parameters can suffer from measurement error and/or high levels of variability due to unknown/unmeasured environmental factors. These effects can counteract genetic effects, the latter providing better proxies for stable long-term exposure of a trait on an outcome.

2. Were UK Biobank MPN cases restricted to participants who had existing diagnoses at the time of blood draw (i.e., initial assessment)? I could not seem to find this information in the Methods. If the ICD10 data

included incident cases, this could be part of the reason that only ~10-17% of individuals had evidence of JAK2 V617F.

Re: We thank the reviewer for this useful comment. The UKBB MPN cases were not restricted to participants who had existing diagnoses at the time of blood sampling. Any participant whose inpatient records can be matched to the MPN ICD10 codes were included as cases in this study.

Following the reviewer's comment, we explored the reasons underlying low *JAK2V617F* positivity in PV. We first considered the time interval from blood draw to diagnosis. To this end, we matched MPN cases to the information about when a blood sample was collected (i.e., date of blood sample collected) and when a particular diagnosis was first recorded in the hospital data (i.e., episode start date) downloaded from the UKBB Data Portal. One MPN case can often be matched to multiple episode start dates and the earliest date was selected as the diagnosis date. We then calculated the time interval between the earliest diagnostic episode and the date of the blood draw across ET and PV cases respectively. We found nearly half of PV (86 out of 161) were diagnosed after sampling, meaning that any somatic mutations in *JAK2* acquired after sampling may not have been present in the blood sample (Figure S9). However, *JAK2V617F* carriers were present in both diagnosis-before-sampling ($n=13$) and diagnosis-after-sampling groups (even more $n=27$), indicating that the time of blood sampling compared to diagnosis would not introduce a strong bias in the analysis regarding the *JAK2V617F*-positive and negative groups.

Supplementary Figure 9 Time difference from blood sample collected to diagnosis in years for ET ($n = 155$) and PV ($n = 161$) patients in UKBB-MPN cohort. Note that the sample size of ET is one less than that in the main analysis ($n = 156$) because of the unavailability of the time of blood sample collected. There were 28 *JAK2V617F* carriers (21 with diagnosis after sampling) across 155 individuals with ET and 40

JAK2V617F carriers (27 with diagnosis after sampling) across 161 individuals with PV (highlighted in yellow).

For PV cases diagnosed before the blood draw, we would expect to see a majority (~95%) to be *JAK2V617F* carriers, however, only 13 (out of 75) were detected by sequencing (**Figure S3**). This could in part be due to therapy that may reduce *JAK2* mutant allele burden, however, we also explored the sequencing coverage and quality of the 200k UKBB-WES data at *JAK2* (which was also requested by Reviewer #1).

The mean number of reads for *JAK2* V617F hotspot was 21.5 (median=21 and s.d.=6.4; **Figure S3e**). The mean coverage of all exons of *JAK2* was 27.5, (median=27 and s.d.=9.6; **Figure S3f**), which is the lowest among all CH genes (see Supplementary Table 1 in Kar *et al.*). This may result in decreased sensitivity of detection of small clones (e.g. VAF <0.1). We now clarify this at the beginning of the Results section (page 3) and add an additional figure (**Figure S3**) as follows: “We studied the germline characteristics of individuals in UK-Biobank (UKBB) with and without *JAK2V617F*. Of 162,534 genetically unrelated individuals with European ancestry within the UKBB whole exome sequencing cohort (200k UKBB-WES as our main discovery dataset; Methods), we identified 540 individuals with one or more mutant reads for *JAK2V617F* (0.3%, median variant allele frequency [VAF] 0.056, range 0.019-1, **Figure S2**; “UKBB-*JAK2V617F* cohort”). The lower rate of *JAK2V617F* in the UKBB-WES, compared to other population studies^{6,7}, could be explained by its low sequencing coverage ($\times 21.5$) as also reported previously¹⁵ (**Figure S3**)”.

The analysis of the pileup output (including read quality for low-quality reads) showed that the distribution of quality scores in the 200k UKBB samples was broadly comparable between *JAK2V617F* positive, *JAK2V617F* negative and other *JAK2* loci (**Figure S3a, b and c**). Furthermore, we did not see a significant difference ($P_{\text{Difference}} = 0.53$) in the coverage between PV-*JAK2V617F* carriers and non-carriers (**Figure S3d**). Therefore, we are reassured that our clinical findings for *JAK2V617F* positive versus negative individuals were not due to differences in sequencing depth or quality of sequencing specifically affecting *JAK2V617F* negative individuals. These data have been added as a new **Figure S3** together with the pile up data above and is shown below.

Supplementary Figure 3 Summary plot for *JAK2V617F* mutation detection in the 200k UK Biobank WES data. Shown in order are the percentages of read quality scores in *JAK2V617F*-positive (a), *JAK2V617F*-negative (b) and other reads (c) in the same exon in *JAK2*. Coverage distribution within PV between *JAK2V617F* carriers and non-carriers is shown in (d) and the distribution of coverage at the V617F hotspot (mean=21.5, median=21, standard deviation [s.d.]=6.4) and all exons of *JAK2* (mean=27.5, median=27, s.d.=9.5) are shown in (e) and (f) respectively, with median values indicated by dash lines. The analysis of the pileup output (including read quality for low-quality reads) showed that the distribution of quality scores in the 200k UKBB samples was broadly comparable between *JAK2V617F* positive, *JAK2V617F* negative and other *JAK2* loci (a, b and c). We did not see a significant difference ($P_{\text{Difference}} = 0.53$) in the coverage between PV-*JAK2V617F* carriers and non-carriers (d).

In the revised manuscript, we described the information of time of diagnosis and blood sampling in the Methods section that “Note that the MPN cases in the UKBB-MPN cohort were not restricted to participants who had existing diagnoses at the time of blood sampling. Any participant whose inpatient records could be matched to the MPN ICD10 codes were included as cases in this study. To calculate the time interval from blood draw to MPN diagnosis, we matched MPN cases to the information about when a blood sample was collected (i.e., date of blood sample collected) and when a particular diagnosis was first recorded in the hospital data (i.e., episode start date) downloaded from the UKBB Data Portal. If an MPN case could be matched to multiple episode start dates, then the earliest date was selected as the diagnosis date. We then calculated the time interval between the two dates across ET and PV cases respectively.” (page 16).

We clarified in the Results section (page 9) that “Of note, only 10-17% of the UKBB-PV cohort had the *JAK2V617F* mutation ($n=1$ or ≥ 2 reads respectively, **Table S1**), when mutated *JAK2* is expected to be found in >99% of PV. This raises the possibility that polygenic germline predisposition to high red blood cell indices may also contribute to other causes of clinical polycythaemia not driven by *JAK2V617F* mutation²⁵. However, we cannot exclude the possibility that some *JAK2* mutations were missed due to the time of blood sampling compared to diagnosis (**Figure S9**), and the low sequencing coverage of *JAK2*, although such factors affected *JAK2V617F*-positive and negative PV subgroups equally (**Figure S3**)”.

We detailed the discussions about the low rate of *JAK2V617F* in the Discussion section (page 14):

We explored the reasons for low *JAK2V617F* positivity in the UKBB-MPN cohort. We found nearly half of the individuals with PV were diagnosed after blood draw in UKBB, potentially explaining the low rate of *JAK2* positivity (**Figure S9**). Furthermore, overall depth of sequencing for *JAK2* was low, as previously reported¹⁵, compared to other CH genes. These factors affected *JAK2V617F* positive and negative PV groups similarly (**Figure S3, S9**) and should not unduly bias the results of germline polygenic risks in the context of somatic mutations in the UKBB-MPN cohort, where a significant differentiation in polygenic effects between top and bottom quintiles of PGS were observed in PV cases who were *JAK2V617F* noncarriers compared to carriers (**Figure 4b**). This would suggest that polygenic germline predisposition contributes to a PV diagnosis in *JAK2V617F*-negative individuals in UKBB and should be considered as a reason for presentation in low-*JAK2*/negative-*JAK2* patients with high haemoglobin or haematocrit.

3. Related to the above observation of unusually low *JAK2-V617F* positivity, the authors "raise the possibility that many individuals in the UKBB-MPN cohort were those with elevated blood counts driven by high PGS" (p.6). This is a testable hypothesis: how do the distributions of the relevant PGSs among UKBB-MPN cases compare to controls?

Re: We thank the reviewer for this suggestion.

As raised by the reviewer, we explained the observation of a “notably higher” contribution of PGS to blood cell traits “in UKBB-MPN than in the MPN-patient cohort (Figure 2d and f)” by the possibility that “many individuals in the UKBB-MPN cohort were those with elevated blood counts driven by high PGS”. We have now rephrased this in the Results section (page 7) that “This may reflect an ascertainment in the estimation of the genetic weights of the PGS from the UKBB, or that some individuals with MPN in the UKBB-MPN cohort are those with elevated blood counts driven by high PGS. This would be consistent with the low prevalence of *JAK2V617F* in the UKBB-MPN cohort compared to the more strictly defined clinical MPN-patient cohort”.

Following the reviewer’s suggestion, we have now directly compared the PGS distribution in the UKBB-MPN cohort to the healthy controls in the UKBB-WES data (200k) for the five identified traits whose PGSs showed a significant predisposition risk to ET/PV (Figure S7). As expected, we see clear case-control shifts for the PGS_{PLT} and PGS_{PCT} in ET and PGS_{HGB}, PGS_{HCT} and PGS_{RBC} in PV.

Extended Figure 1 PGS distributions for the five identified haematological traits between cases in the UKBB-MPN cohort and controls in UKBB (n = 158 ET and 121 PV *JAK2*- vs. 161,872 controls).

This information is already reflected in the significant differentiation of the polygenic risk estimates between the top and bottom PGS quintiles in the MPN cases stratified by somatic mutation (Figure 4). However, we did note the lack of significance in the *JAK2*-positive PV in the MPN-patient cohort (added to Figure 4b as the middle panel). We have discussed above that, although both the low depth of sequencing and time of diagnosis compared to blood draw can contribute to the low rate of *JAK2V617F* in the UKBB-WES, such factors affected *JAK2V617F*-positive and negative groups equally. Therefore, this strengthens the possibility of an inaccurate diagnosis most likely driven by high PGS in the figure caption of Figure 4 (page13):

Figure 4 Germline polygenic risks on ET and PV in the context of somatic mutations. ORs of the top and bottom PGS quintiles relative to that of the middle three quintiles in the MPN-patient cohort (red and yellow) and the UKBB-MPN cohort (blue and grey) for ET (panel **a**) and PV (panel **b**) respectively. Mutation-stratified groups are shown wherever the data are available. Shown *n* is the sample size of ET (or PV) cases in a specific group. It is of note that *JAK2*-negative PV should be a rare diagnosis in PV (normally ~5%) but the number of such patients called in the UKBB-WES data appeared to be high (*n* = 121 out of 161; **Table S1**). It may be that many of these individuals do not have an underlying clonal disorder such as PV but present with secondary erythrocytosis e.g. due to smoking, alcohol, lung disease, where a germline predisposition to high HGB/HCT results in blood counts that mimic PV. However, there was limited statistical power for demonstrating any significance for the *JAK2*+ PV subgroup, given the small sample size (UKBB-MPN cohort, *n* = 40). Furthermore, we cannot exclude the possibility that a mutant *JAK2* clone was missed during sequencing in some of these individuals.

As described above, the germline polygenic risk was quantified based on MPN cases relative to controls considering mutation status only for cases rather than for both. By doing this we included as many controls as possible (regardless of mutation status) and thus maximised our analysis power, but neglected a possibility that those healthy controls may have been exposed to higher disease risk because of carrying MPN driver mutations. Therefore, we further asked the interesting question of whether an individual, even a *JAK2V617F* carrier, can be protected from having MPN by inheriting a low-PGS. For this, we first defined high- and low-PGS groups by the median of PGS distribution for a haematological trait among cases and controls. We then tested if healthy individuals are enriched in the low-PGS group only within *JAK2V617F* carriers for ET and PV separately. We included PGSs of the traits significantly associated with

MPN classification (i.e., HCT, HGB, RBC, PLT and PCT) and also the traits that were causally associated with *JAK2V617F* positivity (i.e., PCT and MONO).

For ET, the estimated OR for PGS_{MONO} is OR = 5.2 [95% CI = 1.7-21.0]; $P = 6.9 \times 10^{-4}$; **Table S12**), indicating that healthy *JAK2V617F* carriers in the low-PGS_{MONO} group are around five times less likely to have ET than that in high-PGS_{MONO} group (interpreted as relative risk which is very close to OR given the low incidence of ET, i.e., 1.60 per 100,000^{8,9}). The effect estimate was also replicated in a logistic regression model (OR = 5.3 with 95% CI = 1.6-17.4 and $P = 0.0054$) with covariates fitted (age, sex, WES batch and PCs).

Supplementary Table 12 Enrichment tests of whether healthy *JAK2V617F* carriers with low PGS (defined by median) were protected from having MPN.

MPN	PGS	#Low PGS		#High PGS		OR (95% CI)	P
		Healthy	MPN cases	Healthy	MPN cases		
ET	HCT	191	14	205	14	0.93 (0.4-2.17)	1.00
	HGB	187	16	209	12	0.67 (0.28-1.56)	0.33
	MONO*	184	4	212	24	5.19 (1.74-20.96)	6.9E-04
	PCT	172	7	224	21	2.3 (0.92-6.56)	0.074
	PLT	192	11	204	17	1.45 (0.62-3.53)	0.43
	RBC	183	12	213	16	1.15 (0.49-2.73)	0.85
PV	HCT	201	17	195	23	1.39 (0.69-2.87)	0.41
	HGB	197	21	199	19	0.9 (0.44-1.81)	0.87
	MONO	193	25	203	15	0.57 (0.27-1.17)	0.13
	PCT	199	19	197	21	1.12 (0.55-2.27)	0.87
	PLT	199	19	197	21	1.12 (0.55-2.27)	0.87
	RBC	198	20	198	20	1 (0.49-2.03)	1.00

*indicates the identified PGS with a significance P -value passed Bonferroni correction ($=0.05/12$).

We update the protective effect of PGS_{MONO} to the Results section (page 8):

We further asked whether an individual, even a *JAK2V617F* carrier, may be protected from developing an MPN by inheriting a low PGS for relevant blood cell traits. By performing enrichment tests across the PGSs of six haematological traits (that were identified to be either putative causal factors for *JAK2V617F* clones or associated factors for MPN diagnosis, i.e., MONO, PCT, PLT, HGB, HCT, RBC; Methods), we found that healthy *JAK2V617F* carriers were enriched in the low-PGS_{MONO} group with enrichment OR = 5.19 [95% CI = 1.74-20.96; $P = 6.9 \times 10^{-4}$], indicating a protective effect making them around five times less likely to have ET than those in the high-PGS_{MONO} group (interpreted as relative risk which is very close to OR given the

low incidence of ET, i.e., 1.60 per 100,000; **Table S12**). Importantly, this indicates that an individual's PGS_{MONO} also influences the risk of developing subsequent disease from *JAK2V617F* CH. The association of low PGS_{MONO} with healthy *JAK2V617F* carriers was also confirmed in a logistic regression analysis (OR = 5.35; 95% CI = 1.64-17.42; $P = 0.0054$) with covariates included (namely age, sex, WES batch and first 10 PCs; Methods).

We add the description of analysis to the Methods section (page 19):

We performed enrichment analysis to test whether there is a polygenic germline protective effect on MPN in *JAK2V617F* carriers in the UKBB-MPN cohort. For this, we first defined high- and low-PGS groups by the median of PGS distribution for a haematological trait among cases and controls. We then isolated *JAK2V617F* carriers and tested if healthy individuals are enriched in the low-PGS group within *JAK2V617F* carriers for ET and PV separately. We considered PGSs for the traits significantly associated with MPN classification (i.e., HCT, HGB, RBC, PLT and PCT) and also the traits that were causally associated with *JAK2V617F* positivity (i.e., PCT and MONO). We also estimated the PGS effect based on a logistic regression model within *JAK2V617F* carriers with healthy carriers coded as cases and carriers with ET (or PV) as controls. The PGS term was fitted as a factor with the low-PGS group (defined by median) coded as

1. $PGS_{MPN-46/1}$, $PGS_{MPN-other}$, PGS_{CH} , age, sex, WES batch and 10 PCs were also fitted as covariates.

4. The definitions of "odds ratios" (OR) reported throughout the manuscript need to be explained more clearly given that the independent variables are usually continuous (e.g., PGSs). Are the PGSs in units of standard deviations of the blood traits, and do the ORs indicate changes in odds per predicted 1 s.d. increase?

Re: The reviewer is correct. PGSs were standardized for each blood trait with units of s.d. and ORs indicate changes in odds per 1 s.d. increase of PGS.

We add a clarification in the Methods section (page 17):

The PGS term in the regression analysis was always standardised with units of s.d.. An estimate of OR indicates changes in odds per s.d. increase in PGS.

We also clarified this at the beginning of the Results section (page 4):

To assess the association between PGSs for haematological traits and small (VAF <0.1, n=397) or large (VAF 0.1 or greater, n=143) *JAK2V617F* clone size, we used multinomial logistic regression including PGSs for each haematological trait (standardised with units of s.d.), and previously reported germline sites associated with MPN⁹ and CH¹⁵ (PGS_{MPN} and PGS_{CH}) as covariates.

5. I found the abstract hard to understand on a first reading, in part because of the meaning of OR and also the mention of "an 'MPN' phenotype" without definition.

Re: We have rephrased the Abstract, including two minor changes for MPN-patient cohort from "OR 1.22, CI=1.01-1.47 to 1.57, CI=1.34-1.86]" to "OR = 1.22 [95% CI = 1.01-1.47] to 1.28 [95% CI = 1.06-

1.55]” and from “OR 1.56, CI=1.44-1.70; OR 1.72, CI=1.44-2.02” to “OR = 1.56 [95% CI = 1.44-1.70] and 1.64 [95% CI = 1.51-1.78]”, because the previous values correspond to UKBB-MPN cohort rather than MPN-patient cohort.

Here is the updated Abstract:

Polygenic germline loci are a major contributor to population variation in common blood cell traits, but little is known about how such predisposition influences clonal selection and cancer. Myeloproliferative neoplasms (MPN) are chronic cancers characterised by excessive production of mature blood cells. Their causative somatic mutations, e.g., *JAK2V617F*, which are common in the population, result in a spectrum of phenotypes from asymptomatic clonal haematopoiesis (CH) to distinct MPN subtypes. We assess how polygenic germline scores (PGSs) based on associated genetic variants for 29 haematological traits drive *JAK2V617F* positivity in UK Biobank (UKBB) and identify PGS for monocyte count (odds ratio [OR] = 1.31 [changes in odds per s.d. increase of PGS], 95% confidence interval [CI] = 1.15-1.49, $P = 4.6 \times 10^{-5}$) and plateletcrit (OR = 1.52, 95% CI = 1.29-1.78; $P = 3.0 \times 10^{-7}$) as new causal risk factors for *JAK2V617F* clonal expansion. Amongst *JAK2V617F* positive individuals, low PGS_{MONO} was enriched for healthy carriers compared to MPN (OR = 5.19, 95% CI = 1.74-20.96, $P = 6.9 \times 10^{-4}$), providing a protective effect making them five times less likely to have MPN compared to

JAK2V617F individuals with high PGS_{MONO}. We genotyped genome-wide SNPs in 761 MPN patients, and built PGSs for blood cell traits in patients and 30,305 healthy controls, and show that germline predisposition to several higher red-cell traits increased the odds of polycythaemia vera (OR = 1.22 [95% CI = 1.01-1.47] to 1.28 [95% CI = 1.06-1.55]), while high PGSs for two platelet traits predisposes to essential thrombocythemia (OR = 1.56 [95% CI = 1.44-1.70] and 1.64 [95% CI = 1.51-1.78]). Results were validated in MPN and healthy controls from UKBB. Importantly, extreme PGSs for blood cell traits may contribute to an MPN diagnosis in the absence of somatic driver mutations in both patients and UKBB, potentially mimicking the phenotype of clonal disease. The heritability of MPN subtype via polygenic predisposition to normal blood traits was independent of known germline MPN and CH risk loci. Our study showcases how polygenic risk underlying common haematological traits influence not only selection on *JAK2V617F* in the healthy population, but also the flavour of subsequent cancer phenotype.

References

1. Cordua, S. *et al.* Prevalence and phenotypes of JAK2 V617F and calreticulin mutations in a Danish general population. *Blood* **134**, 469–479 (2019).
2. Bowden, J. *et al.* Assessing the suitability of summary data for two-sample Mendelian randomization analyses using MR-Egger regression: the role of the I² statistic. *International Journal of Epidemiology* **45**, 1961–1974 (2016).
3. Hardin, J., Schmiediche, H. & Carroll, R. The Simulation Extrapolation Method for Fitting Generalized Linear Models with Additive Measurement Error. *The Stata Journal: Promoting communications on statistics and Stata* **3**, 373–385 (2004).
4. Simulation-Extrapolation Estimation in Parametric Measurement Error Models: Journal of the American Statistical Association: Vol 89, No 428.
<https://www.tandfonline.com/doi/abs/10.1080/01621459.1994.10476871>.
5. Burgess, S. & Thompson, S. G. Interpreting findings from Mendelian randomization using the MR-Egger method. *Eur J Epidemiol* **32**, 377–389 (2017).
6. Ebrahim, S. & Davey Smith, G. Mendelian randomization: can genetic epidemiology help redress the failures of observational epidemiology? *Hum Genet* **123**, 15–33 (2008).
7. Burgess, S., Butterworth, A. S. & Thompson, J. R. Beyond Mendelian randomization: how to interpret evidence of shared genetic predictors. *J Clin Epidemiol* **69**, 208–216 (2016).
8. Zhang, J. & Yu, K. F. What's the Relative Risk? A Method of Correcting the Odds Ratio in Cohort Studies of Common Outcomes. *JAMA* **280**, 1690–1691 (1998).
9. McMullin, M. F. & Anderson, L. A. Aetiology of Myeloproliferative Neoplasms.

Cancers (Basel) **12**, 1810 (2020).

Decision Letter, first revision:

14th September 2023

Dear Jyoti,

Your revised Article "Polygenic germline risk of common haematological traits drives clonal selection on JAK2V617F and the development of myeloproliferative neoplasms" has been seen by the original referees. You will see from their comments below that, while Reviewer #1 has no further requests, Reviewer #2 has requested a few additional analyses and revisions. We remain interested in the possibility of publishing your study in Nature Genetics, but we would like to consider your response to these remaining requests in the form of a further revision before we make a final decision on publication.

As before, to guide the scope of the revisions, the editors discuss the referee reports in detail within the team, including with the chief editor, with a view to identifying key priorities that should be addressed in revision, and sometimes overruling referee requests that are deemed beyond the scope of the current study. In this case, we ask that you perform additional analyses using the full UK Biobank exome release as suggested by Reviewer #2 and revise the presentation taking the results into account. We again hope you will find this prioritized set of referee points to be useful when revising your study. Please do not hesitate to get in touch if you would like to discuss these issues further.

We therefore invite you to revise your manuscript taking into account all reviewer and editor comments. Please highlight all changes in the manuscript text file. At this stage, we will need you to upload a copy of the manuscript in MS Word .docx or similar editable format.

*2) If you have not done so already, please begin to revise your manuscript so that it conforms to our Article format instructions, available

[here](http://www.nature.com/ng/authors/article_types/index.html).

*3) Include a revised version of any required Reporting Summary:

[redacted]

We hope to receive your revised manuscript within 4-8 weeks. If you cannot send it within this time, please let us know.

Nature Genetics is committed to improving transparency in authorship. As part of our efforts in this direction, we are now requesting that all authors identified as 'corresponding author' on published papers create and link their Open Researcher and Contributor Identifier (ORCID) with their account on the Manuscript Tracking System (MTS), prior to acceptance. ORCID helps the scientific community achieve unambiguous attribution of all scholarly contributions. You can create and link your ORCID from the home page of the MTS by clicking on 'Modify my Springer Nature account'. For more information, please visit www.springernature.com/orcid.

Sincerely,
Kyle

Kyle Vogan, PhD
Senior Editor
Nature Genetics
<https://orcid.org/0000-0001-9565-9665>

Referee expertise:

Referee #1: Genetics, cancer, myeloproliferative neoplasms

Referee #2: Genetics, complex traits, clonal hematopoiesis

Reviewers' Comments:

Reviewer #1:
Remarks to the Author:

I share some of the same concerns as Reviewer #2 on the robustness of the MR results, but the authors have performed several diagnostic tests that rule out many of my concerns and have added appropriate text to the Discussion cautioning on interpretation. I recommend the manuscript for publication.

Reviewer #2:
Remarks to the Author:

The authors have done a thorough job addressing my comments, and the revised manuscript is much stronger. I have only two minor follow-up suggestions.

1. Regarding my previous comment 3 (about UKBB-MPN cases being potentially driven by high PGS), I appreciate the authors looking into the PGS distributions in UKBB-MPN cases vs. controls (Extended Figure 1 on p.28 of the response document). I agree that the PGS distributions show case-control shifts as expected, but the bulk of the distributions are actually mostly overlapping, such that it seems to me that at most a few of the MPN cases are likely to be driven by high PGS. The way in which the authors have reworded the main text sounds fine to me, but I suggest also toning down the claim made in the legend of Fig. 4b ("... germline predisposition to high HGB/HCT results in blood counts that mimic PV"). For instance, simply replacing "results in" with "contributes to" would be better. Also, including Extended Figure 1 as a supplementary figure would be helpful to readers -- as I understand, it is currently only provided in the response document.

2. I found the new result on low-PGS_MONO being protective for MPN potentially interesting, but I think this result needs to be evaluated and presented more carefully. The abstract now has a line claiming that JAK2V617F carriers in the low-PGS_MONO group are "five times less likely to have MPN" which I am uncomfortable with for two reasons:

- According to Supplementary Table 12, this result is specific to ET, with PV having the opposite trend: 0.57 (0.27-1.17).
- The confidence interval for ET is very wide (5.19 (1.74-20.96)) and likely reflects some winner's curse given that 12 hypotheses were tested.

Given the wide confidence interval, I strongly recommend rerunning this analysis using data from the full N=470K exome release (which should be very simple given that the authors have now already identified the JAK2V617F carriers) and re-evaluating how to present the results.

Author Rebuttal, first revision:

Response to referees

Referee expertise:

Referee #1: Genetics, cancer, myeloproliferative neoplasms

Referee #2: Genetics, complex traits, clonal hematopoiesis

Reviewers' Comments:

Re: We thank our Reviewers for their feedback on our revised manuscript. Here are our responses to Reviewer #2's follow-up questions. As in the last response letter, we reply to each point in blue and update our manuscript with additional comments/figures/tables in red.

Reviewer #1:

Remarks to the Author:

I share some of the same concerns as Reviewer #2 on the robustness of the MR results, but the authors have performed several diagnostic tests that rule out many of my concerns and have added appropriate text to the Discussion cautioning on interpretation. I recommend the manuscript for publication.

Re: We thank the Reviewer's recognition of our additional analysis and revised manuscript in addressing the concerns on the MR results.

Reviewer #2:

Remarks to the Author:

The authors have done a thorough job addressing my comments, and the revised manuscript is much stronger. I have only two minor follow-up suggestions.

1. Regarding my previous comment 3 (about UKBB-MPN cases being potentially driven by high PGS), I appreciate the authors looking into the PGS distributions in UKBB-MPN cases vs. controls (Extended

Figure 1 on p.28 of the response document). I agree that the PGS distributions show case-control shifts as expected, but the bulk of the distributions are actually mostly overlapping, such that it seems to me that at most a few of the MPN cases are likely to be driven by high PGS. The way in which the authors have reworded the main text sounds fine to me, but I suggest also toning down the claim made in the legend of Fig. 4b ("... germline predisposition to high HGB/HCT results in blood counts that mimic PV"). For instance, simply replacing "results in" with "contributes to" would be better. Also, including Extended Figure 1 as a supplementary figure would be helpful to readers -- as I understand, it is currently only provided in the response document.

Re: We thank the Reviewers for the follow-up suggestion. We have reworded the legend of Figure 4b to tone down the statement as follows: "a germline predisposition to high HGB/HCT may contribute to blood counts phenotypes that mimic PV". We have also added Extended Figure 1 as a supplementary figure (Figure S11) (page 13).

2. I found the new result on low-PGS_MONO being protective for MPN potentially interesting, but I think this result needs to be evaluated and presented more carefully. The abstract now has a line claiming that JAK2V617F carriers in the low-PGS_MONO group are "five times less likely to have MPN" which I am uncomfortable with for two reasons:

- According to Supplementary Table 12, this result is specific to ET, with PV having the opposite trend: 0.57 (0.27-1.17).
- The confidence interval for ET is very wide (5.19 (1.74-20.96)) and likely reflects some winner's curse given that 12 hypotheses were tested.

Given the wide confidence interval, I strongly recommend rerunning this analysis using data from the full N=470K exome release (which should be very simple given that the authors have now already identified the JAK2V617F carriers) and re-evaluating how to present the results.

Re: We thank the Reviewer for their advice and accept their concerns. We have now re-addressed this question in the full UKBB-WES. The new result shows the protective effects of low-PGS on an ET diagnosis for monocytes and two platelet traits with more accurate estimates, while PGS_{MONO} did not pass Bonferroni correction ($P = 0.05/12$; Table S12). We have now updated the results table (Table S12) with the new analysis based on the full UKBB-WES and also the description in the main text (page 8) as follows: "By performing enrichment tests in the full UKBB-WES across the PGSs of six haematological

traits (that were identified to be either putative causal factors for *JAK2V617F* clones or associated factors for MPN diagnosis, i.e., MONO, PCT, PLT, HGB, HCT, RBC; Methods), we found that healthy *JAK2V617F* carriers were enriched in the low-PGS group for the two platelet traits and monocytes, with enrichment OR around 2 (OR = 2.8 [95% CI = 1.51-5.42; $P = 3.810^{-4}$] for PGS_{PCT} , 2.35 [95% CI = 1.29-4.43; $P = 0.0027$] for PGS_{PLT} and 1.99 [95% CI = 1.11-3.67; $P = 0.015$] for PGS_{MONO}), indicating a protective effect making them around two times less likely to have ET than those in the high-PGS group (interpreted as a relative risk, which is very close to OR given the low incidence of ET of ~ 1.6 per 100,000; **Table S12**). Importantly, this indicates that an individual's PGS also influences the risk of developing subsequent disease from *JAK2V617F* CH. The association of low PGS with healthy *JAK2V617F* carriers was also confirmed in a logistic regression analysis (OR = 2.32 [95% CI = 1.27-4.25; $P = 0.0065$] for PGS_{PCT} , 2.48 [95% CI = 1.36-4.54; $P = 0.0032$] for PGS_{PLT} and 2.08 [95% CI = 1.15-3.77; $P = 0.016$] for PGS_{MONO}) with covariates included (namely age, sex, WES batch and first 10 PCs; Methods)."

We also updated the Abstract accordingly:

"Polygenic germline loci are a major contributor to population variation in common blood cell traits, but little is known about how such predisposition influences clonal selection and cancer. Myeloproliferative neoplasms (MPN) are chronic cancers characterised by excessive production of mature blood cells. Their causative somatic mutations, e.g., *JAK2V617F*, which are common in the population, result in a spectrum of phenotypes from asymptomatic clonal haematopoiesis (CH) to distinct MPN subtypes. We assess how polygenic germline scores (PGSs) based on associated genetic variants for 29 haematological traits drive *JAK2V617F* positivity in UK Biobank (UKBB) and identify PGS for monocyte count (odds ratio [OR] = 1.31 [changes in odds per s.d. increase of PGS], 95% confidence interval [CI] = 1.15-1.49, $P = 4.6 \times 10^{-5}$) and plateletcrit (OR = 1.52, 95% CI = 1.29-1.78; $P = 3.0 \times 10^{-7}$) as new causal risk factors for *JAK2V617F* clonal expansion. We genotyped genome-wide SNPs in 761 MPN patients, and built PGSs for blood cell traits in patients and 30,305 healthy controls, and show that germline predisposition to several higher red-cell traits increased the odds of polycythaemia vera (OR = 1.22 [95% CI = 1.01-1.47] to 1.28 [95% CI = 1.06-1.55]), while high PGSs for two platelet traits predisposes to essential thrombocythemia (OR = 1.56 [95% CI = 1.44-1.70] and 1.64 [95% CI = 1.51-1.78]). Results were validated in MPN and healthy controls from UKBB. Amongst *JAK2V617F* positive individuals, low PGS for monocytes and two platelets traits were enriched for healthy carriers (OR = 1.99 [95% CI = 1.11-3.67] to 2.80 [95% CI = 1.51-5.42]), providing a protective effect making them 2-3 times less likely to have essential thrombocythemia compared to *JAK2V617F* individuals with high PGS. Importantly, extreme PGSs for blood cell traits may contribute to an MPN diagnosis in the absence of somatic driver mutations in both patients and UKBB, potentially mimicking the phenotype of clonal disease. The heritability of MPN subtype via polygenic predisposition to normal blood traits was independent of known germline MPN and CH risk loci. Our study showcases how polygenic risk underlying common haematological traits influence not only selection on *JAK2V617F* in the healthy population, but also the flavour of subsequent cancer phenotype."

Supplementary Table 12 Enrichment tests of whether healthy *JAK2V617F* carriers with low PGS (defined by median) were protected from MPN in the full UKBB-WES.

MPN	PGS	#Low PGS		#High PGS		OR (95% CI)	P
		Controls	Cases	Controls	Cases		
ET	HCT	410	33	417	25	0.75 (0.42-1.32)	0.34
	HGB	408	35	419	23	0.64 (0.35-1.14)	0.13
	MONO	423	20	404	38	1.99 (1.11-3.67)	0.015
	PCT*	427	16	400	42	2.80 (1.51-5.42)	3.8E-04
	PLT*	425	18	402	40	2.35 (1.29-4.43)	0.0027
	RBC	417	26	410	32	1.25 (0.71-2.23)	0.42
PV	HCT	412	43	415	40	0.92 (0.57-1.49)	0.82
	HGB	413	42	414	41	0.97 (0.6-1.57)	1.00
	MONO	409	46	418	37	0.79 (0.49-1.27)	0.36
	PCT	416	39	411	44	1.14 (0.71-1.85)	0.65
	PLT	414	41	413	42	1.03 (0.64-1.66)	1.00
	RBC	413	42	414	41	0.97 (0.6-1.57)	1.00

*indicates the identified PGS with a significance *P*-value passed Bonferroni correction ($=0.05/12$).

Decision Letter, second revision:

Our ref: NG-A61494R1

2nd October 2023

Dear Jyoti,

Your revised manuscript "Polygenic germline risk of common haematological traits drives clonal

selection on JAK2V617F and the development of myeloproliferative neoplasms" (NG-A61494R1) has been seen by Reviewer #2. As you will see from the comments below, Reviewer #2 is satisfied with your responses and revisions, and therefore we will be happy in principle to publish your study in Nature Genetics as an Article pending final revisions to comply with our editorial and formatting guidelines.

We are now performing detailed checks on your paper, and we will send you a checklist detailing our editorial and formatting requirements soon. Please do not upload the final materials or make any revisions until you receive this additional information from us.

Thank you again for your interest in Nature Genetics. Please do not hesitate to contact me if you have any questions.

Sincerely,
Kyle

Kyle Vogan, PhD
Senior Editor
Nature Genetics
<https://orcid.org/0000-0001-9565-9665>

Reviewer #2 (Remarks to the Author):

The authors have adequately responded to my follow-up comments, and the updated analyses and presentation of protective effects of low-PGS on MPN are much more reasonable now. Regarding the low-PGS_MONO effect in particular, the authors might want to reconsider highlighting this result in the abstract (given that it failed to replicate among the additional samples and is no longer Bonferroni-significant at $P=0.015$), but I am comfortable now with the overall presentation of these results.

Final Decision Letter: